# Differential negative dominance by *KCNA2* variants associated with global developmental delay suggests *KCNA2* haploinsufficiency in humans

Pei Xin Boon[1] , Amaia Jauregi-Miguel[1] , S. Suheda Yasarbas[1] , Serena Pozzi[1] , Urban Karlsson[1] , Ammar Husami[2] , Charmaine Ko[1] , Amelle Shillington[2] and Antonios Pantazis[1,3] 

[1] *Division of Cell and Neurobiology, Department of Biomedical and Clinical Sciences, Linköping University, Linköping, Sweden*
[2] *Department of Human Genetics, Cincinnati Children's Hospital Medical Center, Cincinnati, OH, USA*
[3] *Wallenberg Center for Molecular Medicine, Linköping University, Linköping, Sweden*

Handling Editors: Eleonora Grandi & Theodore Cummins

The peer review history is available in the Supporting Information section of this article (https://doi.org/10.1113/JP290728#support-information-section).

**Abstract figure legend** Potassium-selective $K_V1.2$ channels are encoded by the *KCNA2* gene and regulate electrical signalling in neurons. *KCNA2* genetic variants are associated with epileptic and developmental encephalopathy. We characterized two variants that subtly alter the channel's amino acid composition, p.H310D and p.G318D. They share several similarities: they occur in proximal and highly conserved positions; they both result in aspartate substitution; and they both prevent the surface-trafficking of $K_V1.2$ subunits, causing loss of *KCNA2* function. Each $K_V1.2$ channel is made of four subunits, and in heterozygous individuals both wild-type and variant subunits coexist and may interact. p.H310D

**Pei Xin Boon** did her doctoral studies in the Smooth Muscle Research Centre at the Dundalk Institute of Technology, Ireland, and graduated with a PhD in Ion Channel Physiology. Her research focused on the investigation of the molecular interaction between the BK channel and its auxiliary subunit, LINGO2. Her interest in ion channels led her to continue as a postdoc in the Pantazis lab. Currently, she utilizes both electrophysiology and immunocytochemical techniques to study the molecular mechanism of disease-associated $K_V1.2$-channel variants and to understand their effects on biosynthesis.

was a strongly negative-dominant variant, as $K_V1.2$(H310D) subunits suppressed the surface trafficking of wild-type $K_V1.2$ subunits. In contrast, $K_V1.2$(G318D) subunits barely influenced the trafficking of wild-type $K_V1.2$ subunits. Since the p.G318D patient nevertheless has neurological symptoms, this suggests that the activity of one wild-type *KCNA2* allele is not enough; that is *KCNA2* may be a haploinsufficient gene in humans.

**Abstract** *KCNA2* encodes the pore-forming subunits of the voltage-gated, potassium-selective channel $K_V1.2$, which controls the excitability of both central and peripheral neurons. Either gain- or loss-of-function *KCNA2* variants can cause severe neurological disease, assigned developmental epileptic encephalopathy (DEE) type 32. Here, we report and characterize two apparently similar variants, p.H310D and p.G318D, both discovered in patients with global developmental delay and involving aspartate substitutions at positions highly conserved in the $K_V$-channel superfamily. We found that both are loss-of-function variants, completely abolishing channel current and subunit trafficking. Channel constructs of $K_V1.2$-variant subunits in tandem with $K_V1.4$ had a conductance with inhibited voltage-dependence, with shifted half-activation potentials by 27 and 19 mV for p.H310D and p.G318D, respectively. p.H310D was strongly negative-dominant: heterozygous cells exhibited only 7% conductance relative to homozygous wild-type, while only half of wild-type sub-units could traffic to the surface. In contrast, p.G318D exhibited weaker negative dominance, with 32% conductance in heterozygous cells and 86% wild-type-subunit trafficking. Taken together with the p.G318D-patient's neurological symptoms, the latter suggests that *KCNA2* is a haploinsufficient gene in humans.

(Received 12 December 2025; accepted after revision 10 March 2026; first published online 30 March 2026)
**Corresponding author** A Pantazis: Linköping University Campus US, Cell Biology, Floor 11, SE-581 85 Linköping, Sweden.    Email: antonios.pantazis@liu.se

## Key points

- *KCNA2* encodes the pore-forming subunits of the $K_V1.2$ voltage-activated, $K^+$-selective ion channel, which regulates electrical signalling in neurons. We characterized two *KCNA2* variants from patients with global developmental delay.
- Both variants are aspartate substitutions of proximal, highly conserved positions in $K_V$-channels: p.H310D and p.G318D.
- In frog oocytes and in primate cells, both variants cause loss of *KCNA2* function, abolishing currents and surface trafficking, and inhibiting channel voltage-dependent opening.
- p.H310D is strongly negative-dominant, potently suppressing wild-type subunit functional expression.
- In contrast, p.G318D is weakly negative-dominant, leaving wild-type subunits largely unaffected. This suggests that *KCNA2* is a haploinsufficient gene in humans.

## Introduction

The first gene associated with epilepsy was identified at the turn of the last century (Steinlein et al., 1995; Oyrer et al., 2018), 2400 years after Hippocrates' stipulation that the 'sacred disease' had a hereditary basis (Temkin, 1933; Wolf, 2025). Among the genes identified to-date and screened in paediatric epilepsy panels (Burk et al., 2024), *KCNA2* is a recent addition, the first reports being published in 2015 (Pena & Coimbra, 2015; Syrbe et al., 2015). *KCNA2* encodes the pore-forming subunits of the tetrameric, delayed rectifier, potassium-selective channel $K_V1.2$, which localizes in neuronal axons and presynaptic terminals, regulating action-potential firing and synaptic release (Trimmer & Rhodes, 2004; Lai & Jan, 2006; Trimmer, 2015).

At the time of writing, 327 missense variants could be found for *KCNA2* in the ClinVar database (Landrum et al., 2018). It is unclear how *KCNA2* variants cause disease; likely it is due to *KCNA2* broad expression patterns in the brain, and the complex interplay of (i) variant effects of neuronal excitability and (ii) neuro-nal network connectivity (Debanne et al., 2024). We do know that both gain- and loss-of-function variants

of *KCNA2* are associated with developmental epileptic encephalopathy (DEE) (Syrbe et al., 2015; Masnada et al., 2017; Doring et al., 2021; Minguez-Vinas et al., 2023). This, together with the broad variation of symptoms and severity, confounds genotype-phenotype correlation and undermines the clinical benefits of broadly adopted genetic screening: while missense variants are continually discovered in patients with DEE, the variant effects on channel function are often uncertain.

Here, we characterized the consequences of two *KCNA2* missense variants, p.H310D and p.G318D. They have several commonalities: (i) both were discovered in patients with global developmental delay, (ii) they occur at proximal sites conserved in the $K_V$-superfamily (positions 310 and 318 in $K_V1.2$), and (iii) both involve aspartate (Asp, D) substitutions. Our objectives were to characterize the consequences of the variants for the channel functional properties and subunit trafficking and to evaluate the interaction of wild-type and variant subunits, emulating the heterozygous condition. We discovered that both variants are loss-of-function and cause complete trafficking deficiency of mutant subunits. p.H310D is also strongly negative-dominant, potently suppressing the surface expression of wild-type sub-units. In contrast, p.G318D is only weakly negative dominant. Combined with the patient's reported neurological phenotypes, the latter finding supports the premise that *KCNA2* is a haploinsufficient gene in humans.

## Methods

### Ethical approval

The individual described here is a minor. The individual's parents have provided written permission to participate in research and publication, under CCHMC IRB-2013-7327. The study conformed to the principles of the *Declaration of Helsinki*, except for registration in a public database. The use of *Xenopus laevis* animals, including the performed surgery, was reviewed and approved by the regional board of ethics in Linköping, Sweden (case no. 14515).

Defolliculated *X. laevis* oocytes were either purchased from Ecocyte (Dortmund, Germany) or isolated and prepared from locally kept frogs supplied by Nasco (Fort Atkinson, WI, USA). Up to five female frogs were kept in an aquarium with dimensions 615 mm × 435 mm × 232 mm (length, breadth, height) at the Linköping University animal facility. Aquarium water was cleansed continuously using mechanical and biological filters, and 10–15% of the water was changed weekly. Water temperature was kept at 17–19°C, and conductivity at 400–1000 µS. Frogs were fed twice per week. Artificial mangrove roots and a glass container with stones and large limestone were lowered into the aquariums for environmental enrichment, shelter and as an additional source of lime. Animal-facility staff monitored the frogs daily. The facilities are approved by the Swedish Agricultural Agency (Jordbruksverket).

To collect ovaries, frogs were anaesthetized in a water bath containing 1.4 g/l MS-222 Sandoz, 2.4 g/l 4-(2-hydroxyethyl)-1-piperazineethanesulfonic acid (HEPES), pH adjusted to 7.5 with 10 M NaOH. Anaesthesia was confirmed by absence of reflex responses to foot pinch. The anaesthetized frog was placed on ice and her abdomen was cleaned with 70% ethanol. Lobes of ovaries were removed through a 15-mm abdominal incision and placed into $Ca^{2+}$-free OR-2 solution (in mM: 82.5 NaCl, 2.5 KCl, 1 $MgCl_2$ and 5 HEPES; pH adjusted to 7.4 by NaOH) for further processing. Frogs were treated with analgesics (5 mg/ml Marcain and 2% Xylocain) and the incision was sutured. Frogs were immersed in water with a pillow under the chin, to keep their nose above water, and observed until regaining consciousness (ca. 30 min) prior to being returned to a recovery aquarium for post-surgical monitoring. Frogs were allowed to recover for at least 2 months between surgeries, which alternated between left and right side of the abdomen. Frogs were humanely euthanized by decapitation under deep anaesthesia (by placing in a water bath containing 1.4 g/l MS-222 Sandoz, 2.4 g/l HEPES, pH adjusted to 7.5 with 10 M NaOH; anaesthesia was confirmed by absence of reflex responses to foot pinch) if any of these conditions were met: (i) they were sick or not thriving; (ii) they had undergone six surgeries, or (iii) they were housed at the animal facility for over 5 years. The investigators understand the ethical principles under which the journal operates and that their work complies with *The Journal of Physiology*'s animal ethics checklist.

### Patient clinical genetic testing

Clinical exome sequencing was performed at Cincinnati Children's Hospital Medical Centre on genomic DNA using the Human Comprehensive Exome kit from Twist Bioscience (South San Francisco, CA, USA) to enrich the whole exome. The exome was sequenced using an Illumina (San Diego, CA, USA) sequencing system with paired-end reads at a minimum coverage of 20× of 95% of the target regions. This individual's exome DNA sequences were aligned to the human reference genome (build UCSC hg19) with BWA-mem. Variants were called using GATK, and QC was performed as part of an in-house-developed pipeline based on GATK best practices. Sequencing was completed for the proband along with parental samples as a trio study. This resulted in a variant of interest in *KCNA2*: Chr1(GRCh37):g.111146452C>T ENST00000485317:c.953G>A (p.Gly318Asp).

## Molecular biology

pcDNA3 plasmids containing human $K_V1.2$ with an N-terminally fused enhanced green fluorescent protein (EGFP) or a monomeric red fluorescent protein (mRFP1) tag and an extracellular haemagglutinin (HA) site (between transmembrane helices S1 and S2, as applicable) were used; these constructs were synthesized as described previously (Nilsson et al., 2022). Site-directed mutagenesis was performed using high-fidelity *Pfu* polymerase (Agilent Technologies, Santa Clara, CA, USA, 600850) or Q5-high fidelity DNA polymerase (New England Biolabs, Ipswich, MA, USA, M0491S). Plasmid purification was performed using the ZymoPURE Plasmid Miniprep and DNA Clean & Concentrator Kits (Zymo Research, Irvine, CA, USA). The rest of the biological reagents were provided by New England Biolabs and all synthetic oligonucleotides were synthesized by Integrated DNA Technologies (Leuven, Belgium), unless stated otherwise. All molecular operations were confirmed by full sequencing at the Molecular Biology Unit at the Linköping University Core Facility. For electrophysiology experiments, pMAX plasmids were linearized using *Pac*I and transcribed into cRNA using *in vitro* transcription (messageMAX T7, Cellscript, Madison, WI, USA). cRNA was purified using Monarch RNA Cleanup Kit (New England Biolabs), quantified spectrophotometrically, evaluated by gel electrophoresis and stored at −80°C. All constructs are available upon reasonable request.

## Surface-trafficking assays

COS-7 cells (Merck; ECACC 87021302) were grown in Complete Culture Medium containing Dulbecco's modified Eagle's medium (DMEM)/F12 Nutrient Mixture (1:1) (Thermo Fisher Scientific, Waltham, MA, USA), 10% heat-inactivated fetal bovine serum (FBS), 100 U/ml penicillin, 100 µg/ml streptomycin and 0.5 mM glutamine. Cultures were incubated at 37°C with 5% $CO_2$ and passaged twice to thrice per week up to passage 22. Cells were plated in 12-well plates (approximately 150,000 cells per well) and transfected within 24 h after seeding using jetOPTIMUS transfection reagent (Polyplus) using 1 µl jetOPTIMUS and 1 µg plasmid DNA (i.e., 0.5 µg for each construct in biallelic assays).

To evaluate the surface protein expression of $K_V1.2$ subunits, COS-7 cells were collected by trypsinization (0.05%) 48 h post transfection. Cells were washed with Dulbecco's phosphate buffered saline (DPBS, Thermo Fisher Scientific) and pelleted at 400 *g* for 5 min at 4°C. Cells were stained with Brilliant Violet 421 anti-HA.11 Epitope Tag (8 µg/ml, BioLegend, San Diego, CA, USA) in 100 µl DPBS/5% FBS for 45 min at 4°C. Cells were washed with (i) DPBS / 5% FBS and (ii) DPBS. After the final wash, cells were resuspended in 300 µl DPBS. Finally, a single-cell suspension was prepared using pre-separation filters (Miltenyl Biotec, Bergisch Gladbach, Germany, 70 µm) and stored at 4°C in the dark until measurement. Cells were measured using a FACSAria III Cell Sorter with FACSDiva 8.0.2 software (BD Biosciences, San Jose, CA, USA) at the Flow Cytometry Unit of the Linköping University Core Facility. Brilliant Violet 421, EGFP and mRFP1 were excited at 405, 488 and 553 nm, respectively. The emission filters were 450/40, 550/30 and 660/20 nm. All gates were set as in our previous studies using the same experimental paradigm (Nilsson et al., 2022; Minguez-Vinas et al., 2023); briefly, cell aggregates and debris were excluded based on forward and side scatter; fluorescence gates were defined using negative controls; and spectral overlap between channels was corrected using single-colour control samples. The minimum cell count for EGFP- or mRFP1-positive cells was 5000. For biallelic assays, only cells positive for both EGFP and mRFP1 signals were included in the analysis for each replicate.

## Electrophysiology

Defolliculated *X. laevis* oocytes were either purchased from Ecocyte or isolated and prepared from locally kept frogs as previously described (Minguez-Vinas et al., 2023). Briefly, retrieved ovaries were cut into small clusters, subsequently treated with Liberase (7 units/batch; Roche, Mannheim, Germany, 05401127001) in ∼10 ml of OR-2 (in mM, 82.5 NaCl, 2.5 KCl, 1 $MgCl_2$, 5 HEPES; pH adjusted to 7.0 by NaOH) with agitation using an orbital shaker for 30–40 min. Liberase was removed by washing with OR-2 solution, followed by 30 min agitation to remove the follicular layers. Mature (V–VI) defolliculated oocytes were selected and stored at 17°C in standard oocyte solution (SOS; in mM, 100 NaCl, 2 KCl, 1.8 $CaCl_2$, 1 $MgCl_2$ and 5 HEPES; pH adjusted to 7.0 by NaOH), supplemented with penicillin (100 units/ml), streptomycin (100 µg/ml) and gentamicin (50 µg/ml).

Each oocyte was injected with 50 nl containing: (homozygous wild-type or mutant condition) 6 ng of wild-type or mutant $hK_V1.2$ cRNA; (half-dose wild-type) 3 ng of wild-type $hK_V1.2$ cRNA; (heterozygous) 3 ng wild-type $hK_V1.2$ cRNA and 3 ng mutant $hK_V1.2$ cRNA (1:1) or 3 ng wild-type $hK_V1.2$ cRNA and 6 ng mutant $hK_V1.2$ cRNA (1:2). $K_V1.4/K_V1.2$ concatemer cRNA was injected at 3.5 ng per oocyte, 1 day before experiments. All experiments were completed with 2–3 batches of oocytes, except the $K_V1.2$-$K_V1.4$ experiment (Fig. 4), where one batch was investigated per variant.

Cut-open oocyte Vaseline gap (COVG) voltage-clamp (Taglialatela et al., 1992; Stefani & Bezanilla, 1998; Pantazis & Olcese, 2019) experiments were as previously described (Pantazis et al., 2020; Nilsson et al., 2022; Minguez-Vinas

et al., 2023). External solution (mM): 120 sodium methanesulfonate (MeS), 10 K-MeS, 2 Ca(MeS)$_2$, 10 HEPES (pH 7.0); internal solution (mM): 120 potassium glutamate, 10 HEPES (pH 7.0); intracellular micropipette solution (mM): 2700 Na-MeS, 10 NaCl. Oocytes were injected with cRNA 2 days before experimentation.

Two-electrode voltage clamp (TEVC) recordings were performed in an OpusXpress 6000A (Axon Instruments, Union City, CA, USA) as described previously (Nilsson et al., 2022), unless stated otherwise (see below). SOS was used as external solution. Oocytes were injected with cRNA 1 day before experimentation.

Niflumic acid (NFA) experiments were performed using a TEVC set-up with the Geneclamp 500B amplifier (Molecular Devices, San Jose, CA, USA). NFA stock was prepared in dimethyl sulfoxide (100 mM). Oocytes were first mounted in SOS containing 0.3% dimethyl sulfoxide as vehicle control, and subjected to a standard current-voltage (*I-V*) protocol. During NFA perfusion (300 µM), current was recorded at –20 mV; a second *I-V* protocol was executed when the current had reached steady-state, which was invariably within 5 min of the start of perfusion.

### Trafficking assay data analysis

Data were processed and quantified using Kaluza Analysis (Beckman Coulter Life Sciences, Indianapolis, IN, USA). Cell surface staining was normalized ($x_{norm}$) using the following formula:

$$x_{norm} = \frac{x - \overline{x_0}}{\overline{x_{WT}} - \overline{x_0}}$$

where $x$ is the mean, $\log_{10}$ anti-HA measurement from live, EFP-positive (or EGFP- and mRFP1-positive, as applicable) cells, $\overline{x_0}$ is the mean of mean, $\log_{10}$ anti-HA measurements from live, mRFP1-positive cells transfected with mRFP1-K$_V$1.2 (without a HA tag) from the same batch, and $\overline{x_{WT}}$ is the mean of mean, $\log_{10}$ anti-HA measurements from live, EGFP-positive (or EGFP- and mRFP1-positive, as applicable) cells transfected with EGFP-K$_V$1.2(HA) from the same batch. The mean of negative control (mRFP1-K$_V$1.2) is 0 and EGFP-K$_V$1.2(HA) is 1.

**Electrophysiology analysis.** All curve fitting was performed by least squares using *Solver* in Microsoft Excel 365, unless stated otherwise. First, macroscopic conductance (*G*) was calculated by dividing the current ($I_m$) measured at the end of the test pulse by the driving force:

$$G = \frac{I_m}{V_m - E_K}$$

where $E_K$ is the equilibrium potential for potassium. For fast-inactivating currents (from tandem concatemers with N-terminal K$_V$1.4), the peak current was measured. Steady-state activation was then calculated by fitting *G* to a Boltzmann distribution with one or two terms (*i*):

$$G = G_{max} \sum_{i=1}^{2} \frac{w_i}{1 + \exp\left[\frac{z_i F}{RT}\left(V_{0.5,i} - V_m\right)\right]}$$

where $V_m$ is the membrane potential, $G_{max}$ is the maximal conductance, $V_{0.5}$ is the half-activation potential, $z$ is the effective valence, F and R are the Faraday and gas constants, respectively, and $T$ is the temperature (294 K). $w_1$ was limited to values from 0 to 1, and $w_2$ was constrained to be $1 - w_1$. When using the sum of two Boltzmann distributions, the effective $V_{0.5}$ was computed iteratively using *Solver* in Microsoft Excel 365.

**Protein structures.** The structure of rat K$_V$1.2 in the open, conducting state (EMD-43134, PDB: 8VC6) (Wu et al., 2025) was rendered using UCSF ChimeraX (Pettersen et al., 2021).

**Statistics.** All values are reported to two significant figures, except *P*-values, which are reported to three significant figures. All measurements were taken from distinct samples. All comparative statistics (e.g. relative $G_{max}$) were obtained among oocytes or COS-7 cells from the same batch and injection or transfection, as applicable. *P*-values were calculated using an unpaired two-sample permutation-based Student's *t* test (*permuttest2*, part of Permutools; Crosse et al., 2024) in MATLAB R2024b (MathWorks, Natick, MA, USA) with use of Welch's *t*-statistic for samples of unequal size or variance and max correction of family-wise error rate (FWER). Errors and error bars are reported as standard deviation (SD).

## Results

### Background and clinical presentation of the variants

p.H310D (NM_004974.4(KCNA2):c.928C>G) was reported in ClinVar (variation ID: 1801703; accession: VCV001801703.1) as a germline mutation from a patient with global developmental delay. The histidine at K$_V$1.2 position 310 is conserved (Fig. 1) (Minguez-Vinas et al., 2023) and present in 27 members out of 40 in the voltage-gated, potassium-selective (K$_V$) channel family. Accordingly, MetaDome (Wiel et al., 2019) assigned the p.H310D variant a score of 0.12, 'highly intolerant'. Located at the cytosolic flank of transmembrane helix S4, it is among the last amino acids of the voltage-sensor domain (Fig. 1): a conserved protein module that undergoes conformational changes upon membrane depolarization, conferring to ion channels and enzymes

the ability to sense electrical signals (Borjesson & Elinder, 2008; Chanda & Bezanilla, 2008; Jan, 2025). We previously investigated *KCNA2* variants p.H310R and p.H310Y, which surprisingly altered $K_V1.2$ functional expression in opposite directions: the former was a loss-of-function variant, while the latter increased channel function (Minguez-Vinas et al., 2023). Of the 40 $K_V$-channel genes, 27 possess a histidine at this position and 10 have been reported to harbour disease-associated variants, according to ClinVar.

The p.G318D patient is female and was 10 years old at the time of writing, with muscle weakness, global developmental delay, obstructive sleep apnoea, duplicated left kidney collection system and laryngeal cleft. Her history includes febrile seizures and two generalized tonic–clonic seizures. She also has cortical vision abnormalities, attention-deficit/hyperactivity disorder, staring spells, and borderline intellectual disability (intelligence quotient 76). The variant is *de novo* (Fig. S1). She also has Charcot–Marie–Tooth disease type 1A (CMT1A), due to an inherited duplication of the *PMP22* gene (Roa et al., 1993). However, CMT1A does not present this early, and the duplication does not explain the congenital or central nervous system abnormalities, so the *KCNA2* variant is proposed to underlie the early-onset,

central nervous system pathology—as has been reported in other patients with *KCNA2* variants (Syrbe et al., 2015; Masnada et al., 2017; Doring et al., 2021). Additional clinical information about the patient is provided in the online Appendix. Like p.H310D, p.G318D is located at a conserved site: a glycine encountered in 36 out of 40 $K_V$-family members, at the middle of the intracellular S4–S5 linker (Fig. 1). MetaDome assigned this variant a 0.11 'highly intolerant' score. Substitution of the homologous position in the Shaker $K_V$ channel from *Drosophila* by a fluorescent unnatural amino-acid (G386Anap) resulted in defective channel opening (Kalstrup & Blunck, 2018). In ClinVar, one can find disease-associated variants reported in 10 $K_V$-channel genes at this position11 counting also *KCNA2*. The S4–S5 linker connects the voltage sensor to the channel pore domain, and transduces voltage-dependent conformational changes to the gate, that is, the part of the channel that determines open probability (Blunck & Batulan, 2012).

## Both variants are trafficking-deficient

We first sought to evaluate the variant effects on basic channel functional properties. However, cells injected with cRNA encoding human $K_V1.2$(H310D)

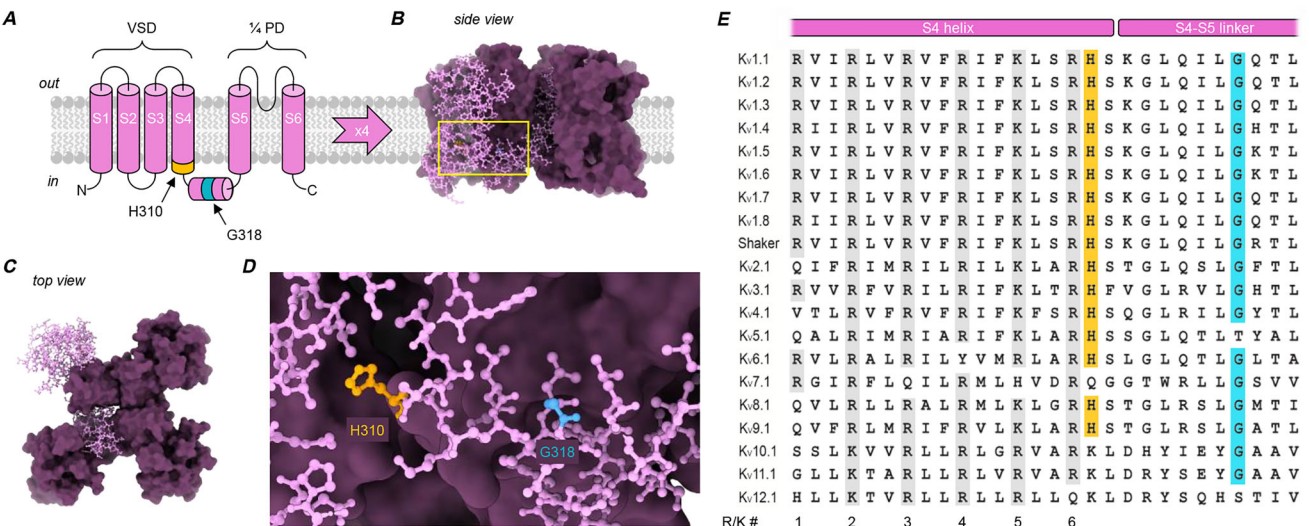

**Figure 1. H310 and G318 are proximal positions in $K_V1.2$ channels and conserved throughout the $K_V$-channel superfamily**

*A*, membrane topology of a $K_V1.2$ subunit, the protein product of the *KCNA2* gene. The subunit possesses six transmembrane segments (S1–S6). S1–S4 comprise a voltage-sensor domain (VSD) and S5–S6 contribute to the central, potassium-selective pore domain (PD). H310 (orange) and G318 (blue) are located at the cytosolic flank of S4, and the S4–S5 linker, respectively. *B*, a single $K_V1.2$ subunit is shown in ball-and-stick depiction, embedded in a homo-tetrameric channel (structure resolved in Wu et al., 2025). *C*, top view of the channel in panel *B* showing how the central PD is surrounded by four peripheral VSDs. *D*, close-up of the area bounded by the yellow box in panel *B*, showing H310 and G318. *E*, amino-acid sequence alignment of the S4 and S4–S5 linker regions in select $K_V$-channel subunits (Minguez-Vinas et al., 2023). All sequences are human, except Shaker, the archetypal $K_V$-channel from *Drosophila melanogaster*. Positively charged, voltage-sensing residues in S4 are highlighted grey and numbered below. H310 and G318 are highlighted orange and blue, respectively.

or $K_V1.2$(G138D) exhibited no significant macroscopic current, even in extreme depolarizations of 200 mV (Fig. 2*A* and *B*), indicating loss of function. In cells transfected with EGFP- and HA-tagged $K_V1.2$ constructs engineered to evaluate surface trafficking (Fig. 2*C*) (Gu et al., 2003; Pantazis et al., 2020; Nilsson et al., 2022; Minguez-Vinas et al., 2023), both substitutions completely prevented the trafficking of mutant subunits to the cell surface (Fig. 2*D* and *E*). Our interpretation of the negative normalized surface-staining values for H310D and G318D (Fig. 2*E*) is that the negative control construct (mRFP1-tagged $K_V1.2$ without a HA tag), which should reach the cell surface, receives some non-specific $\alpha$-HA staining.

## Both variants inhibit voltage-dependent opening

To gain a glimpse of the variant effects on channel function, we sought to rescue the functional expression of variant $K_V1.2$ subunits by fusing their cDNA with that of $K_V1.4$ (*KCNA4*). Relative to other members of the $K_V1$-channel family, $K_V1.4$ subunits are more competent at trafficking and do co-assemble with $K_V1.2$-subunits *in vivo*, to form hetero-tetrameric channels with diverse functional properties (Li et al., 2000; Manganas et al., 2001; Misonou & Trimmer, 2004). Fusing $K_V1.2$ and $K_V1.4$ cDNA results in concatenated channel subunits that assemble as 'dimers of dimers' and form pseudotetrameric channels (Fig. 3*A*); in this way, trafficking-deficient variant subunits $K_V1.2$(F233S) and $K_V1.2$(H310R) were

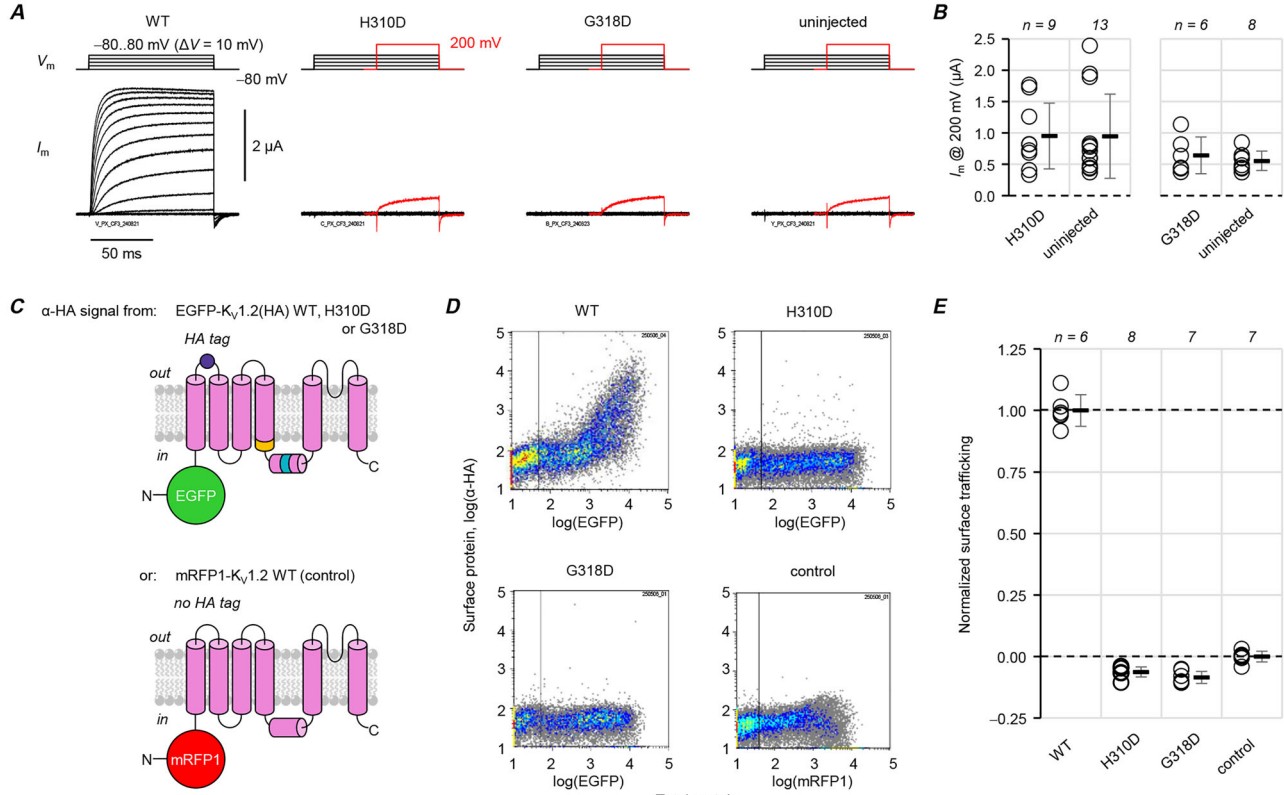

**Figure 2. p.H310D and p.G318D are loss-of-function, trafficking-deficient variants**
*A*, representative COVG records from oocytes injected with human wild-type (WT) $K_V1.2$, $K_V1.2$(H310D), or $K_V1.2$(G318D) cRNA, and from uninjected cells. *B*, cells injected with variant cRNA exhibited current comparable to that of uninjected cells (measured at 200 mV). Left, H310D: $0.95 \pm 0.52$ µA; uninjected: $0.95 \pm 0.67$ µA, $P = 0.906$. Right, G318D: $0.64 \pm 0.29$ µA; uninjected: $0.56 \pm 0.15$ µA, $P = 0.808$. *C*, constructs to evaluate $K_V1.2$ cell surface trafficking. Top, $K_V1.2$ with N-terminally fused EGFP, reporting total protein production; and an extracellular haemagglutinin (HA) tag. Bottom, as negative control, an mRFP1-fused, wild-type $K_V1.2$ was used, without a HA tag. *D*, representative cell density plots from flow cytometry (FC) surface-trafficking assays. Each dot represents a single, unpermeabilized COS-7 cell transfected with the constructs from panel *C*. The *x*-axis indicates total subunit, from the N-terminally fused fluorescent protein. The *y*-axis indicates fluorescence from an antibody against the HA tag. The vertical line separates cells negative (left) and positive (right) for EGFP or mRFP1 signal. *E*, normalized $\alpha$-HA signals from EGFP- or mRFP1-positive cells show that H310D and G318D subunits do not reach the cell surface. WT: $100 \pm 6.4\%$; H310D: $-6.3 \pm 2.0\%$; G318D: $-8.5 \pm 2.4\%$; control: $0.0 \pm 2.2\%$. Errors are SD.

rescued when they were expressed in tandem with a $K_V1.4$ subunit (Nilsson et al., 2022; Minguez-Vinas et al., 2023).

Cells injected with cRNA from the $K_V1.4$-$K_V1.2$ wild-type construct exhibited fast-inactivating currents (Fig. 3*B*), due to the free, N-terminal, $K_V1.4$ inactivation particle in the construct (Tseng-Crank et al., 1993; Nilsson et al., 2022; Minguez-Vinas et al., 2023; Sukomon et al., 2023; Tan et al., 2025). Cells with either $K_V1.4$-$K_V1.2$(H310D) or $K_V1.4$-$K_V1.2$(G318D)

constructs also showed fast-inactivating currents, indicating functional rescue. They differed from cells expressing $K_V1.4$-$K_V1.2$(wt) channels in two seminal ways: (i) they had significantly less conductance (~17%, Fig. 3*C*) and (ii) their currents had significantly more depolarized voltage dependence (H310D: $\Delta V_{0.5} = 27$ mV; G318D: $\Delta V_{0.5} = 19$ mV; Fig. 3*D*). A decrease in overall conductance can be interpreted as either less surface trafficking or decreased ability of the channel to open. The right-shifted voltage dependencies mean that more

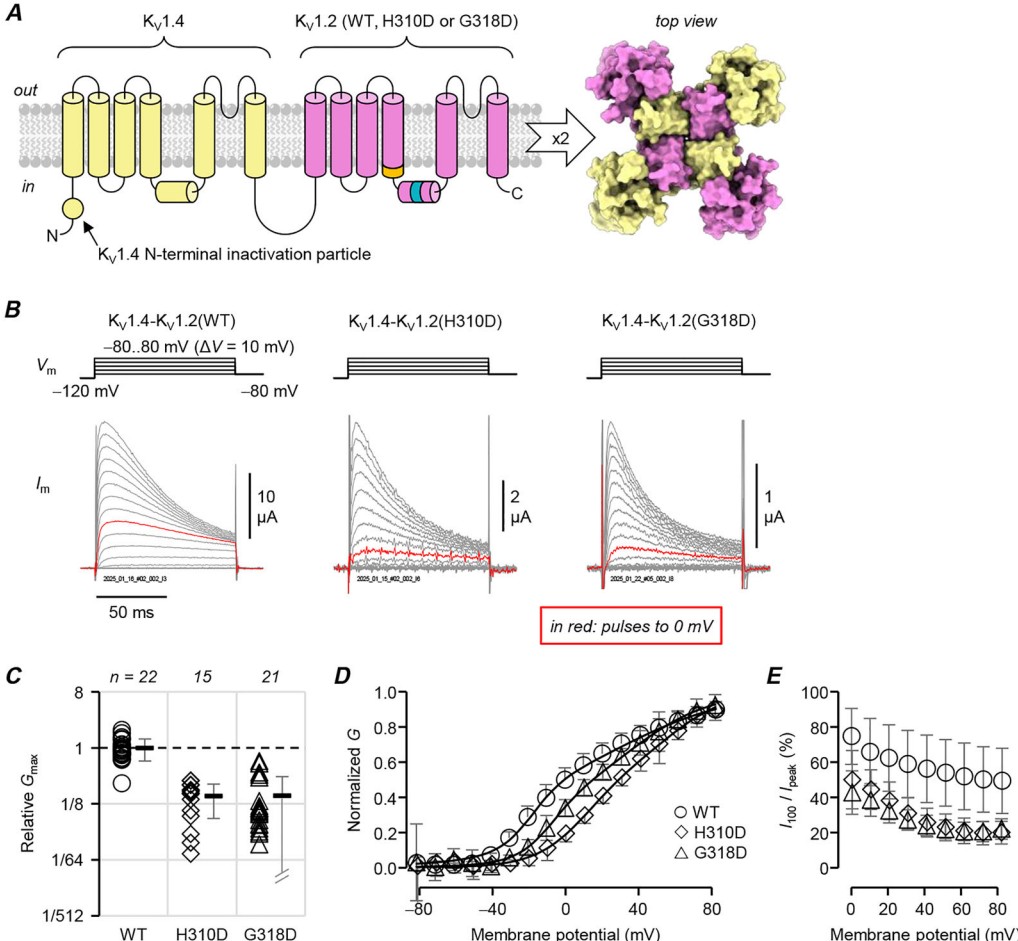

**Figure 3. p.H310D and p.G318D inhibit voltage-dependent opening**
*A*, $K_V1.2$ variants were fused into tandem constructs with N-terminal $K_V1.4$ subunits, to study their effects on channel voltage-dependent opening in 'dimers of dimers' channels (Nilsson et al., 2022; Minguez-Vinas et al., 2023). Note that $K_V1.4$ subunits have an N-terminal inactivation particle (Tseng-Crank et al., 1993; Sukomon et al., 2023; Tan et al., 2025). *B*, representative TEVC records from oocytes injected with $K_V1.4$-$K_V1.2$, $K_V1.4$-$K_V1.2$(H310D) and $K_V1.4$-$K_V1.2$(G318D) cRNA. Note the fast current inactivation. The traces were normalized to be the same height, and the trace in red shows the current at 0 mV. *C*, total peak conductance ($G_{max}$) normalized to $K_V1.4$-$K_V1.2$ (○, $100 \pm 39$%). $K_V1.4$-$K_V1.2$(H310D) (◇): $17 \pm 9.7$%, $P = 0.00160$; $K_V1.4$-$K_V1.2$(G318D) (△): $17 \pm 18$%, $P = 0.00190$. *D*, mean, normalized peak conductance (*G*) plotted against the test potential and fit to the sum of two Boltzmann distributions, from the same cells as in panel *C*. $K_V1.4$-$K_V1.2$ (○; $V_{0.5} = -3.7 \pm 8.3$ mV; $\Delta V_{0.5} = 0.0 \pm 3.9$ mV); $K_V1.4$-$K_V1.2$(H310D) (◇; $V_{0.5} = 31 \pm 4.5$ mV; $\Delta V_{0.5} = 27 \pm 13$ mV; $P = 0.0103$) and $K_V1.4$-$K_V1.2$(G318D) (△; $V_{0.5} = 17 \pm 9.0$ mV; $\Delta V_{0.5} = 23 \pm 9.1$ mV; $P = 0.0148$). *E*, current at the end of the 100 ms pulse ($I_{100}$) over peak current ($I_{peak}$) plotted against the test potential. $I_{100}/I_{peak}$ values for the pulse to 80 mV: $K_V1.4$-$K_V1.2$ (○): $49 \pm 19$%; $K_V1.4$-$K_V1.2$(H310D) (◇): $20 \pm 6.5$% ($P = 0.414$) and $K_V1.4$-$K_V1.2$(G318D) (△): $22 \pm 5.8$% ($P = 0.370$). Errors are SD.

depolarization is required to open these channels, that is, voltage-dependent opening is inhibited.

The variants did not seem to affect the action of the $K_V1.4$ inactivation particle: while there was a tendency for the constructs with variant $K_V1.2$ to inactivate more (compared to the $K_V1.4$-$K_V1.2$ wild-type construct), the effect was not significant (Fig. 3*E*).

When the $K_V1.2$-variant cDNA was positioned at the N-terminus of the concatenated construct, neither construct produced detectable current (Fig. 4). This result, that $K_V1.2$-$K_V1.4$ dimeric constructs exhibit less functional expression when the $K_V1.2$-variant partner is N-terminal, was also observed in other trafficking-deficient variants (p.F233S, p.H310R) (Nilsson et al., 2022; Minguez-Vinas et al., 2023). Post-translational modifications, like phosphorylation and glycosylation, can affect $K_V1.2$- and $K_V1.4$-subunit trafficking (Li et al., 2000; Watanabe et al., 2004; Yang et al., 2007), although these are presumably preserved when $K_V1.2$ is either N- or C-terminal. Our result here suggests that (i) a trafficking defect occurs early in biosynthesis, for example, during translation or endoplasmic-reticulum translocation; and (ii) constructs with an N-terminal $K_V1.4$ partner may traffic better due to cooperative translocation.

### p.H310D is strongly negative-dominant

Since both variants caused loss of function, we sought to determine their effect on the functional expression of wild-type $K_V1.2$ subunits. The oocyte expression system allows the precise dosing of wild-type and variant cRNA, with single-cell precision, to evaluate the current exhibited by 'homozygous wild-type' cells (injected with a 'double' dose, or 6 ng per oocyte, of wild-type cRNA); 'single-dose' wild-type (3 ng wild-type cRNA per oocyte); and 'heterozygous' (3 ng wild-type and 3 ng variant cRNA). Cells heterozygous for p.H310D exhibited 7.3% macroscopic conductance relative to homozygous wild-type (Fig. 5*A* and *B*). This residual conductance had virtually identical voltage-dependence to that of wild-type $K_V1.2$ (Fig. 5*C*), likely from channels comprising only wild-type subunits, $K_V1.2$(wt), that evaded association with $K_V1.2$(H310D)-subunits in the cell.

To confirm that $K_V1.2$(wt)-subunit trafficking was impaired in the presence of $K_V1.2$(H310D), we performed a 'bi-allelic' surface trafficking assay (Fig. 5*D*). In this experiment, the trafficking of EGFP-$K_V1.2$(HA, wt) subunits was evaluated in cells positive for mRFP1-$K_V1.2$, wild-type or H310D. The mRFP1 construct represented the second allele and did not bear a HA tag. In the presence of mRFP1-$K_V1.2$(H310D), only half of the EGFP-$K_V1.2$(HA, wt) subunits reached the surface (49%), compared to in the presence of mRFP1-$K_V1.2$(wt) (100%) (Fig. 5*E* and *F*). By associating with H310D-subunits

during biosynthesis, wild-type subunits were sequestered in the cell: a mechanism of negative dominance (Nilsson et al., 2022).

If $K_V1.2$(wt) and $K_V1.2$(H310D) subunits could interact, would this lead to some rescue of H310D-subunits? In the presence of mRFP1-$K_V1.2$(wt), EGFP-$K_V1.2$(HA,H310D) subunits exhibited 5.4% trafficking (Fig. 5*E*,*F*), at the limit of reliable detection: it was not significantly different from in the presence of mRFP1-$K_V1.2$(H310D) ($P = 0.159$), or from the negative staining control ($P = 0.644$).

### p.G318D exhibits weaker negative dominance than p.H310D

Given their similarities so far, we expected p.G318D to fully recapitulate the properties of p.H310D. However, p.G318D-heterozygous cells exhibited $31 \pm 7.3\%$ conductance relative to homozygous wild-type (Fig. 6*A* and *B*); this was substantially higher than the relative conductance of p.H310D-heterozygous cells (7.3%, Fig. 5*A* and *B*), and not significantly different from the conductance of 'single-dose' wild-type cells (59%, $P = 0.526$; Fig. 6*A* and *B*). We tested for this striking result in an alternative, fully blinded experiment, comparing conductance between the 'single-dose' wild-type, heterozygous p.H310D and heterozygous p.G318D conditions (Fig. 6*D–F*). Compared to the former (100%), heterozygous p.H310D cells exhibited 25% conductance, while heterozygous p.G318D cells exhibited 52% conductance. This was in agreement with the experiment in Fig. 6*A* and *B*, whereby heterozygous cells exhibited roughly half the conductance of cells with a single dose of wild-type RNA. Similar to p.H310D, the conductance of heterozygous cells had similar voltage-dependence to wild-type (Fig. 6*C* and *F*).

The weakness of p.G318D negative dominance became clearer in the bi-allelic trafficking assays (Fig. 6*G–I*). EGFP-$K_V1.2$(HA, wt)-subunit trafficking was impaired in the presence of mRFP1-$K_V1.2$(G318D), albeit very mildly (86%). No rescue of mutant subunit trafficking was detected in the presence of wild-type subunits.

### Effects of a KV1.2-channel opener

Niflumic acid (NFA) facilitates $K_V1.2$-channel voltage-dependent opening (Servettini et al., 2023), so we sought to evaluate its ability to augment the currents of cells in the heterozygous condition. Figure 7 shows that NFA perfusion (300 µM) recapitulated the previously reported augmentation of $K_V1.2$ opening, shifting voltage dependence by *ca.* 10 mV to more negative voltages (Fig. 7*D–F*). Looking more closely at the physiologically relevant pulses to −20 mV, currents were increased

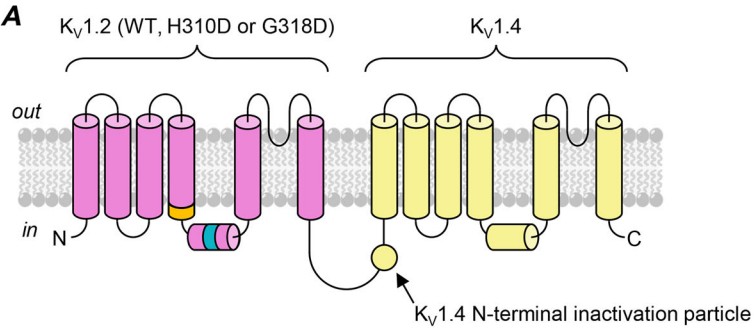

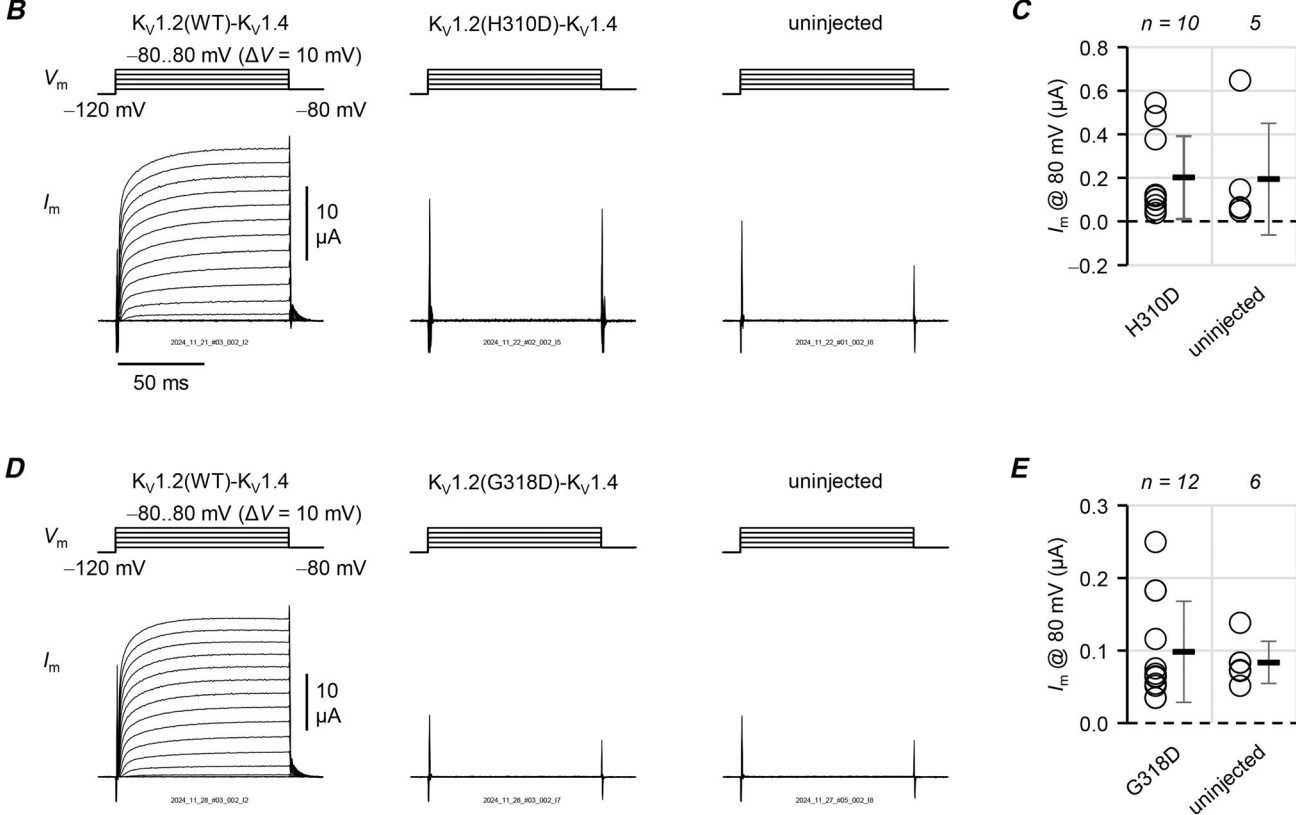

**Figure 4. N-terminal variant K$_V$1.2-subunits prevent functional expression in K$_V$1.2-K$_V$1.4 tandem constructs**

*A*, K$_V$1.2 variants were fused into tandem constructs with C-terminal K$_V$1.4 subunits, to study their effects on channel voltage-dependent opening in 'dimers of dimers' channels (Nilsson et al., 2022; Minguez-Vinas et al., 2023). Note that the K$_V$1.4 N-terminal inactivation particle is now tethered by the linker connecting the subunits. *B*, representative TEVC records from oocytes injected with K$_V$1.2-K$_V$1.4 or K$_V$1.2(H310D)-K$_V$1.4 cRNA, and uninjected oocytes. Thus, in the construct with wild-type K$_V$1.2, the current does not exhibit fast inactivation, as observed previously (Nilsson et al., 2022; Minguez-Vinas et al., 2023). No current was produced by the K$_V$1.2(H310D)-K$_V$1.4 construct, in contrast to the construct where K$_V$1.4 was the N-terminal partner (Fig. 3). *C*, current measured upon depolarization to 80 mV. K$_V$1.2(H310D)-K$_V$1.4: 0.20 ± 0.19 μA. Uninjected: 0.19 ± 0.26 μA; *P* = 0.967. *D* and *E*, as in panels *B* and *C*, respectively, for the G138D variant. K$_V$1.2(G318D)-K$_V$1.4: 0.098 ± 0.070 μA. Uninjected: 0.084 ± 0.029 μA; *P* = 0.947. Errors are SD.

by ∼3-fold, for all tested conditions (wild-type and heterozygous p.H310D and p.G318D) (Fig. 7G).

## Discussion

### Variant effects

Here, we investigated two *KCNA2* variants associated with global developmental delay, p.H310D and p.G318D. Both were determined to have loss-of-function effect, completely preventing the surface trafficking of variant subunits and eliminating potassium conductance (Fig. 2). The variants also showed negative dominance, suppressing the conductance and surface-trafficking of wild-type subunits, albeit to different extents, p.H310D being notably more negative-dominant (Fig. 5) than p.G318D, which was on the threshold of no negative dominance (Fig. 6). We can assert that the strong negative dominance of p.H310D emerges from its ability to interact with wild-type subunits, reducing their surface trafficking (Fig. 5E and F). The weaker negative dominance of p.G318D likely arises from a decreased ability of the mutant subunits to associate with other subunits, which also explains the trafficking deficiency. The S4–S5 linker, where G318 is located, abuts the neighbouring subunit in the channel tetramer

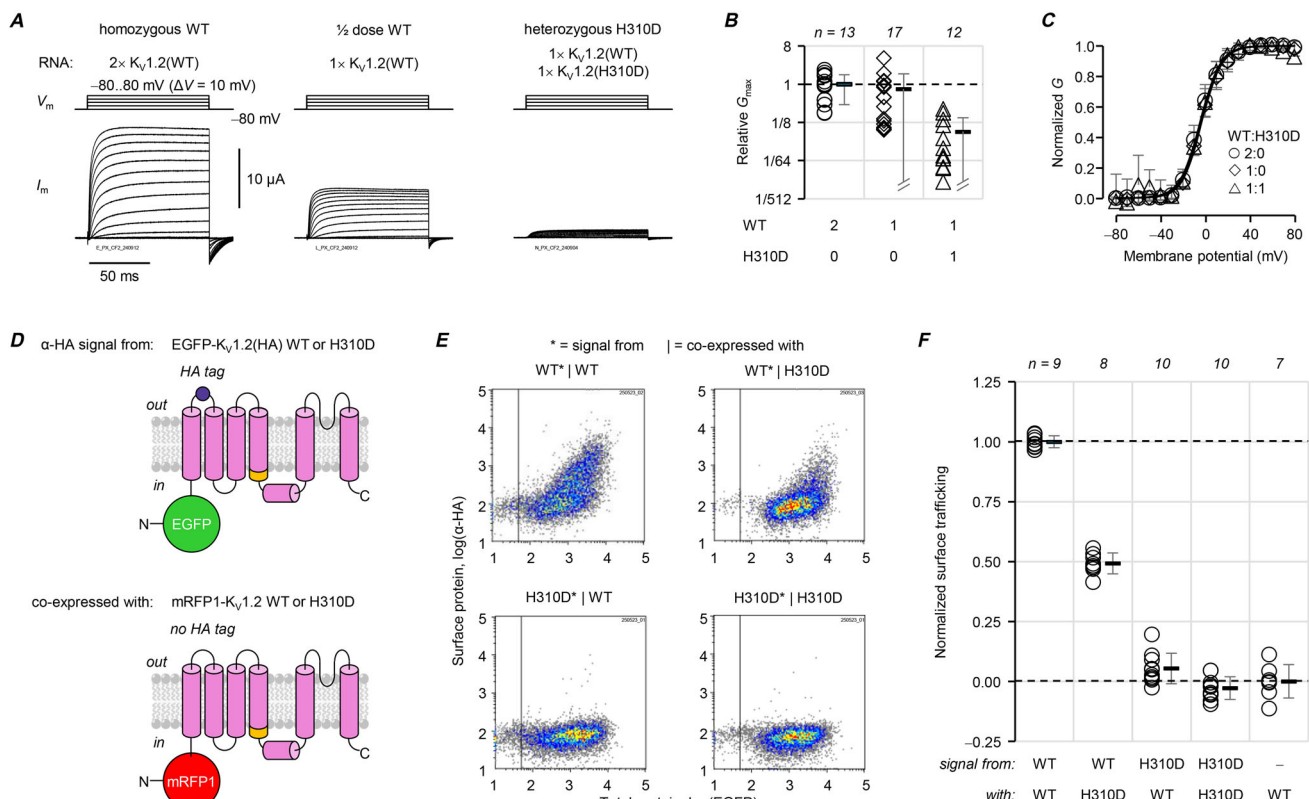

**Figure 5. p.H310D is strongly negative dominant**

*A*, representative COVG records from oocytes injected with 'double-dose' (2×) human wild-type (WT) $K_V1.2$ cRNA, emulating the homozygous wild-type condition; 'single-dose' (1×) WT cRNA; and equimolar wild-type and p.H310D cRNA, emulating the heterozygous condition. *B*, maximal conductance ($G_{max}$) normalized to homozygous (2×) wild-type (○, 100 ± 67%). 1×WT (◇): 77 ± 100%; heterozygous (△): 7.3 ± 8.8%. *C*, mean, normalized conductance (*G*) plotted against the test potential and fit to the Boltzmann distribution, from the same cells as in panel *B*. Homozygous wild-type (○; $V_{0.5} = -4.7 \pm 3.5$ mV; $z = 3.1 \pm 0.76 \, e_0$), 1×WT (◇; $V_{0.5} = -2.9 \pm 3.5$ mV; $z = 2.9 \pm 0.58 \, e_0$) and heterozygous (△; $V_{0.5} = -3.8 \pm 3.8$ mV; $z = 3.3 \pm 0.80 \, e_0$). *D*, constructs for bi-allelic trafficking assays. Top, HA-tagged $K_V1.2$ with N-terminally-fused EGFP. Bottom, mRFP1-fused $K_V1.2$ without a HA tag. *E*, representative cell density plots. These show surface (α-HA) signal (*) from cells transfected by the EGFP-$K_V1.2$, HA-tagged, construct, which are also gated for (|) a mRFP1-$K_V1.2$ construct without a HA tag. For example, the notation 'WT*|H310D' means the trafficking of EGFP-$K_V1.2$(HA) subunits in cells positive for the mRFP1 signal from co-transfected mRFP1-$K_V1.2$(H310D). *F*, normalized α-HA signals from EGFP- and mRFP1-positive cells show that wild-type subunit trafficking is strongly suppressed in the presence of $K_V1.2$(H310D) subunits: WT*|WT: 100 ± 2.5%; WT*|H310D: 49 ± 4.3%; $P = 6.00 \times 10^{-4}$. Moreover, $K_V1.2$(H310D) subunits are not rescued by wild-type: H310D*|WT: 5.4 ± 6.4%; compared to H310D*|H310D: −2.8 ± 4.8%; $P = 0.159$; or control: 0.0 ± 7.0%, $P = 0.644$. Errors are SD.

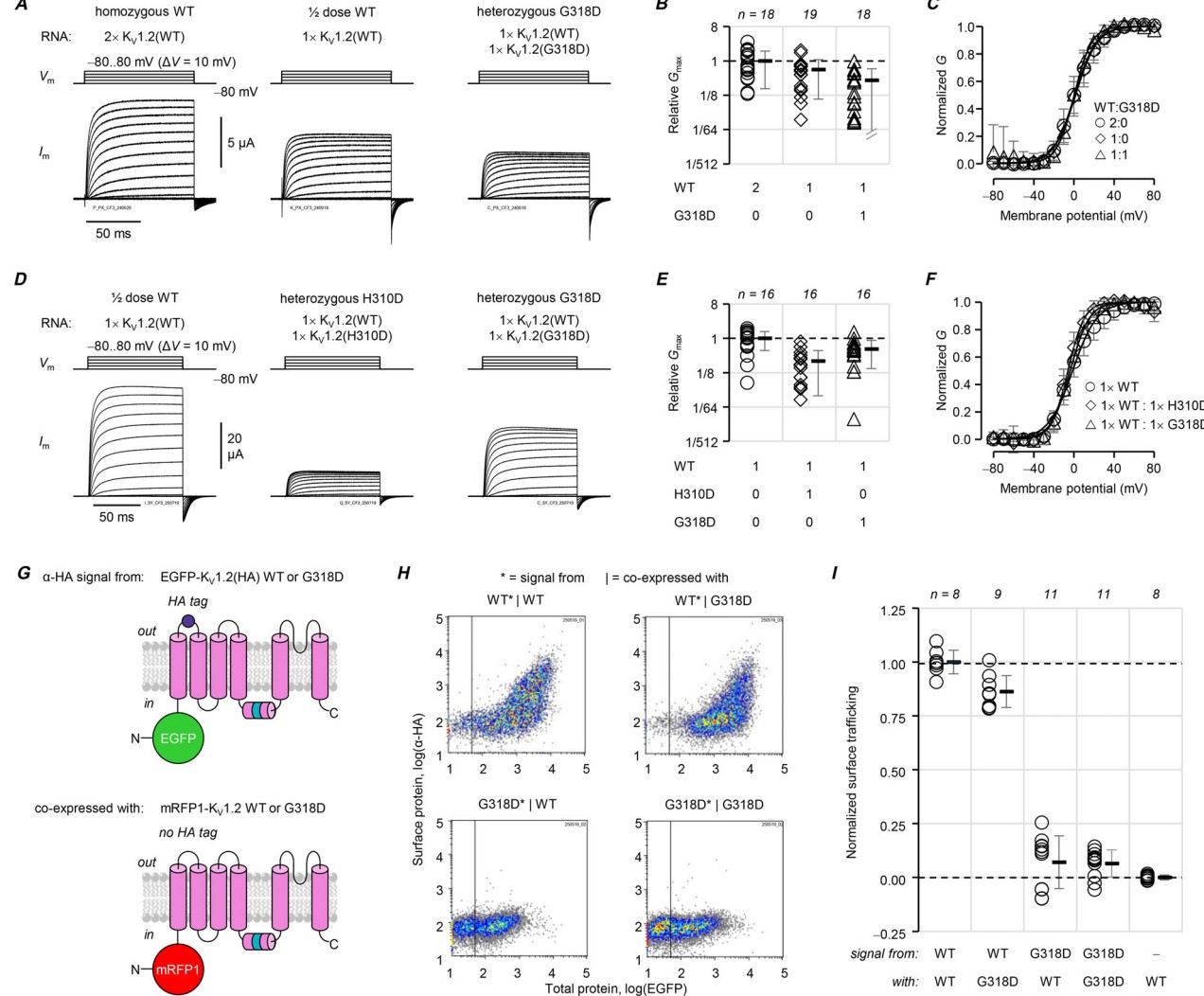

**Figure 6. p.G318D is weakly negative-dominant**

*A*, representative COVG records from oocytes injected with 'double-dose' (2×) human wild-type (WT) $K_V 1.2$ cRNA, emulating the homozygous wild-type condition; 'single-dose' (1×) WT cRNA, and equimolar wild-type and p.G318D cRNA, emulating the heterozygous condition. *B*, maximal conductance ($G_{max}$) normalized to homozygous (2×) wild-type (○, 100 ± 81%). 1×WT (◇): 59 ± 49%; heterozygous (△): 31 ± 31%. N.b., *P* for 1×WT and heterozygous was 0.526. *C*, mean, normalized conductance (*G*) plotted against the test potential and fit to the Boltzmann distribution, from the same cells as in panel *B*. Homozygous wild-type (○; $V_{0.5} = 0.98 ± 5.8$ mV; $z = 2.6 ± 0.36\ e_0$), 1×WT (◇; $V_{0.5} = 1.6 ± 6.1$ mV; $z = 2.6 ± 0.36\ e_0$) and heterozygous (△; $V_{0.5} = -0.032 ± 2.9$ mV; $z = 2.9 ± 0.48\ e_0$). *D*, representative COVG records from a different set of blinded experiments, with oocytes injected with 'single-dose' (1×) human wild-type (WT) $K_V 1.2$ cRNA, equimolar wild-type and p.H310D, or equimolar wild-type and p.H310D cRNA, emulating the heterozygous conditions. *E*, maximal conductance ($G_{max}$) normalized to 1× wild-type (○, 100 ± 52%). Heterozygous p.H310D (◇): 25 ± 22%, $P = 1.00 × 10^{-4}$. Heterozygous p.G318D (◇): 52 ± 36%, $P = 0.00945$. *F*, mean, normalized conductance (*G*) plotted against the test potential and fit to the Boltzmann distribution, from the same cells as in panel *B*. 1× wild-type (○): $V_{0.5} = -0.59 ± 5.9$ mV, $z = 2.2 ± 0.65\ e_0$. Heterozygous H310D (◇): $V_{0.5} = -5.6 ± 4.5$ mV; $z = 3.3 ± 0.53\ e_0$). Heterozygous G318D (△): $V_{0.5} = -3.0 ± 4.5$ mV; $z = 2.8 ± 0.53\ e_0$). For these experiments, blinding was used during the preparation of cRNA injection samples, the experiments and their analysis. *G*, constructs for bi-allelic trafficking assays. Top, HA-tagged $K_V 1.2$ with N-terminally fused EGFP. Bottom, mRFP1-fused $K_V 1.2$ without a HA tag. *H*, representative cell density plots. These show surface ($\alpha$-HA) signal (*) from cells transfected by an EGFP-$K_V 1.2$, HA-tagged, construct, that are also gated for (|) a mRFP1-$K_V 1.2$ construct without a HA tag. For example, the notation 'WT*|G318D' means the trafficking of EGFP-$K_V 1.2$(HA) subunits in cells positive for mRFP1-$K_V 1.2$(G318D) subunits. *I*, normalized $\alpha$-HA signals from EGFP- and mRFP1-positive cells show that wild-type subunit trafficking was not significantly suppressed in the presence of $K_V 1.2$(G318D) subunits: WT*|WT: 100 ± 5.5%; WT*|G318D: 86 ± 7.4%; $P = 0.106$. Moreover, $K_V 1.2$(G318D) subunits were not rescued by wild-type: G318D*|WT: 7.1 ± 12%; compared to G318D*|G318D: 6.5 ± 6.3%; $P = 1.00$; or negative control: 0.0 ± 1.1%; $P = 0.790$. Errors are SD.

(Fig. 1*D*) (Wu et al., 2025), so misfolding at that location may impair assembly.

Successful rescue of the variant subunits in concatemers with $K_V1.4$ showed that they also inhibited channel voltage-dependent activation (Fig. 3). In this way, the subunits recapitulate the 'dual loss of function' property of $K_V1.2$-channel variants p.F233S (Nilsson et al., 2022) and p.H310R (Minguez-Vinas et al., 2023). A rightward shift in the voltage dependence of channel opening suggests that either channel active/open states are destabilized, or that resting/closed states are more stable, relative to the wild-type. Here, the constructs produced channels comprising two $K_V1.4$ and two $K_V1.2$-variant subunits (Fig. 3*A*). Voltage-dependence defects in subunits can have an additive effect to the tetrameric-channel voltage-dependence (Gagnon & Bezanilla, 2010). For this reason, we speculate that if $K_V1.2$-variant homo-tetramers could traffic to the cell surface, they would exhibit double the $V_{0.5}$-shifts relative to wild-type, that is, 54 mV

for p.H310D and 38 mV for p.G318D. This would almost completely diminish their opening at physiological membrane voltage.

Our work on heterologous expression systems in the absence of neuronal $K_V1.2$ molecular partners is too removed from the context of the human brain for a definitive discussion on epilepsy and developmental delay. We speculate that loss of $K_V1.2$ function would broaden the action potentials of both excitatory and inhibitory neurons. In excitatory synapses, this would increase synaptic release (Kole et al., 2007). Yet in inhibitory neurons, fast-gating $K_V3$ channels (Sekirnjak et al., 1997; Erisir et al., 1999; Tasic et al., 2016) would act to normalize action potential duration (Rowan et al., 2014; Labro et al., 2015); in this way, inhibitory synaptic release would be preserved. The resulting imbalance of excitatory and inhibitory transmission could thus promote epileptogenesis (McCormick & Contreras, 2001).

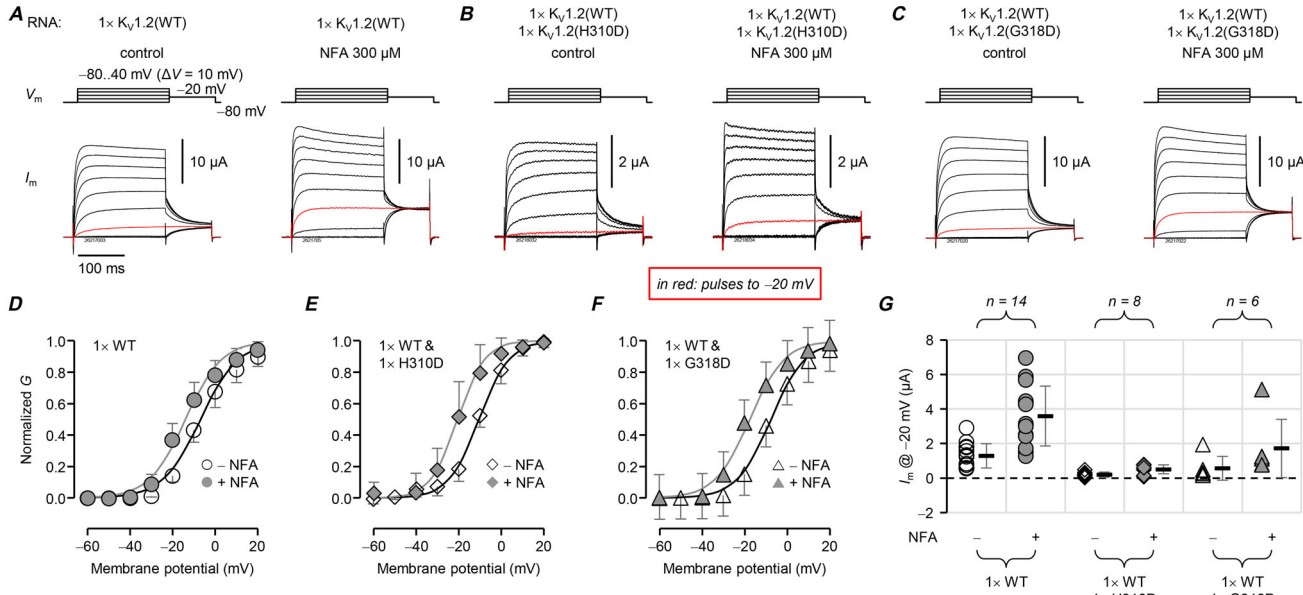

**Figure 7. NFA increases $K_V1.2$ current in heterozygous cells**
*A*, representative TEVC records from oocytes injected with 'single-dose' (1×) human wild-type $K_V1.2$ cRNA, before (left) and after (right) perfusion of 300 μM niflumic acid (NFA). Control is vehicle control. Traces in red are from the −20 mV pulse. *B* and *C*, as in panel *A*, from cells injected with equimolar wild-type and H310D, or G318D, $K_V1.2$ cRNA, respectively. *D*, mean, normalized conductance (*G*) plotted against the test potential and fit to the Boltzmann distribution, from cells expressing wild-type $K_V1.2$. Before NFA (○): $G_{max} = 0.11 \pm 0.046$ mS; $V_{0.5} = -5.5 \pm 3.6$ mV; $z = 2.9 \pm 0.76$ $e_0$. After NFA 300 μM (●): $G_{max} = 0.12 \pm 0.048$ mS; $V_{0.5} = -13 \pm 4.7$ mV; $z = 2.9 \pm 0.68$ $e_0$; $n = 14$ cells. *E*, mean, normalized conductance (*G*) plotted against the test potential and fit to the Boltzmann distribution, from cells expressing equimolar wild-type and H310D $K_V1.2$. Before NFA (◇): $G_{max} = 0.014 \pm 0.0088$ mS; $V_{0.5} = -11 \pm 2.6$ mV; $z = 3.7 \pm 0.85$ $e_0$. After NFA 300 μM (◆): $G_{max} = 0.014 \pm 0.0097$ mS; $V_{0.5} = -19 \pm 5.9$ mV; $z = 4.0 \pm 1.1$ $e_0$; $n = 8$ cells. *F*, mean, normalized conductance (*G*) plotted against the test potential and fit to the Boltzmann distribution, from cells expressing equimolar wild-type and G318D $K_V1.2$. Before NFA (△): $G_{max} = 0.042 \pm 0.049$ mS; $V_{0.5} = -7.5 \pm 2.3$ mV; $z = 3.3 \pm 0.70$ $e_0$. After NFA 300 μM (▲): $G_{max} = 0.046 \pm 0.050$ mS; $V_{0.5} = -18 \pm 4.4$ mV; $z = 3.4 \pm 0.65$ $e_0$; $n = 6$ cells. *G*, current at −20 mV. Wild-type, before NFA (○): $1.3 \pm 0.70$ μA; after NFA 300 μM (●): $3.6 \pm 1.7$ μA. Wild-type and H310D, before NFA (◇): $0.20 \pm 0.15$ μA; after NFA 300 μM (◆): $0.50 \pm 0.26$ μA. Wild-type and G318D, before NFA (△): $0.56 \pm 0.69$ μA; after NFA 300 μM (▲): $1.7 \pm 1.7$ μA. Errors are SD.

## A hierarchy among the negative-dominant variants

Similar to the two variants studied here, loss-of-function variant p.F233S also exhibits negative dominance due to a trafficking defect: heterozygous cells exhibit ~20% conductance relative to homozygous wild-type (Nilsson et al., 2022), placing it at an intermediate level of negative dominance between p.H310D (7% heterozygous conductance; Fig. 5*B*) and p.G318D (31% heterozygous conductance; Fig. 6*B*). This pattern holds for the extent of wild-type-subunit trafficking in the presence of variant subunits. p.H310D has the strongest effect (49% wild-type trafficking, Fig. 5*F*), followed by p.F233S (55%) (Nilsson et al., 2022) and p.G318D (86%, Fig. 6*I*). Thus, p.H310D is the strongest dominant-negative *KCNA2* variant reported to-date, while the weak negative dominance of p.G318D is of particular importance as it suggests that *KCNA2* is a haploinsufficient gene in humans – further discussed below.

## Haploinsufficiency in *KCNA2*

By the measures discussed above, p.G318D appears to be the least negative-dominant, loss-of-function variant of *KCNA2*. In heterozygous cells, the surface trafficking of wild-type subunits and overall potassium conductance is at the limit of a theoretical heterozygous condition. However, the patient has notable neurological symptoms. Combined, these findings support the premise that *KCNA2* is haploinsufficient in humans. Yet while it is promising that the two variants showed differential negative dominance in diverse cellular contexts and culture conditions, we cannot exclude that p.G318D exhibits strong negative dominance in neurons, in the developing human brain. Testing the impact of the p.G318D variant on trafficking in neurons would be an important step to support or disprove this premise.

Haploinsufficiency would not have been possible to evaluate in other loss-of-function variants, which generally exhibit negative dominance—likely because of the obligatory tetrameric assembly of $K_V$ channels that results in wild-type/variant-subunit hetero-tetramers, with decreased ability to traffic and/or function (Nilsson et al., 2022). In addition, trafficking-positive variants present several confounding effects, due to (i) the multiple functions of $K_V$1.2 channels (e.g., opening/closing, inactivation onset and recovery, facilitation; Rezadeh et al., 2007) that may be differentially affected by mutations (Masnada et al., 2017; Pantazis et al., 2020; Minguez-Vinas et al., 2023); (ii) the ability of $K_V$1.2 sub-units to associate with other members of the $K_V$1-channel family, resulting in multiple channel configurations with varied functional properties (Manganas & Trimmer, 2000); and (iii) the broad expression pattern of *KCNA2* in neurons with different $K_V$1.2-activity regulators (Baronas

et al., 2018; Lamothe et al., 2024), diverse excitability properties (Trimmer & Rhodes, 2004; Lai & Jan, 2006) and distinct contributions of circuit function. p.G318D effectively acts as a null mutation, completely preventing the trafficking of variant $K_V$1.2-subunits to the cell surface, or their rescue by wild-type subunits (Fig. 6*H* and *I*). This property simplifies this discussion, as we need not consider effects of channels with altered biophysical properties, on the excitability of diverse neuronal types.

*Kcna2* heterozygous (+/−) mice showed subtle signs of haploinsufficiency (Brew et al., 2007): they lived as long as their homozygous wild-type littermates, contrary to their *Kcna*-null littermates, which all perished by P19 (Brew et al., 2007). The authors wrote 'adult +/− mice were good breeders, suggesting that they flourish despite the substantially reduced expression of Kcna2 mRNA and $K_V$1.2 protein demonstrated at P14' (Brew et al., 2007). The deaths of null mice were attributed to severe generalized spontaneous seizures, which were not reported in the heterozygous mice (Brew et al., 2007). Nevertheless, heterozygous mice did show some neurological differences relative to homozygous wild-type: (i) they showed intermediate susceptibility to flurothyl-evoked seizures, and (ii) their auditory neurons of the medial nucleus of the trapezoid body (MNTB) were hypoexcitable (Brew et al., 2007). It is not very clear that the mouse heterozygous phenotype matches in severity with that of the human p.G318D patient. In another study, mice with region-specific *Kcna2* hetero-knockout in the CA3 hippocampal subfield exhibited haploinsufficiency with respect to threshold for long-term potentiation and were 'impaired in discrimination of similar but slightly distinct contexts' (Eom et al., 2022). Human heterozygous *KCNA2* knock-out induced pluripotent stem cells were recently generated (Shan et al., 2024), promising human-specific insights.

A major advantage of a diploid genome is haplotype redundancy; so why would an ion-channel gene be haploinsufficient? According to the hybrid model for haploinsufficiency, haploinsufficient genes are under pressure to produce only a narrow range of protein activity, which is evidenced by the negative outcomes of their overexpression (Morrill & Amon, 2019). The existence of pathogenic *KCNA2* gain-of-function variants (Syrbe et al., 2015; Masnada et al., 2017; Hedrich et al., 2021; Imbrici et al., 2021; Minguez-Vinas et al., 2023) is in line with *KCNA2* being haploinsufficient according to this model. The same pattern seems to follow for related gene *KCNA1*, which was reported to be haploinsufficient (Zerr et al., 1998) and has gain-of-function variants (e.g. Muller et al., 2023). However, in a different study, *KCNA1* was reported to have recessive inheritance (Verdura et al., 2020). It will be interesting to determine whether haplotype insufficiency is a conserved property for the *KCNA* ($K_V$1) family, or specific to *KCNA2*.

The clinical significance of *KCNA2* haploinsufficiency is on the efficacy of treatments using anti-sense oligonucleotides (ASOs), aiming to abolish negative-dominant subunits by silencing their RNA (Huang et al., 2024). By eliminating variant transcripts, ASO treatments could theoretically restore $K_V1.2$ activity up to 50% relative to the homozygous wild-type condition, depending on ASO potency (to remove variant RNA), and specificity (for variant over wild-type RNA). Yet if *KCNA2* were haploinsufficient, restoring up to 50% activity would not be enough to prevent deleterious effects. Bearing in mind the caveat that p.G318D may be more negative-dominant in the human brain than in our experiments, we propose that the addition of $K_V1$-channel openers (e.g. Borjesson et al., 2008; Liin et al., 2018; Silvera Ejneby et al., 2020; Manville et al., 2023; Servettini et al., 2023; Ronnelid & Elinder, 2024; Manville et al., 2025a; Manville et al., 2025b) in a combined ASO/pharmaceutical strategy, is likely to produce the most efficacious therapeutic outcome.

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

## Additional information

### Data availability statement

All data are available upon reasonable request from the corresponding author. Permutools (Crosse et al., 2024) was downloaded from: https://github.com/mickcrosse/PERMUTOOLS

### Competing interests

The authors declare they have no competing interests.

## Author contributions

A.S. contributed the clinical description. P.X.B, S.S.Y, S.P. and C.K. performed experiments. P.X.B, A.J.M., S.S.Y, S.P. and C.K. performed analysis. P.X.B., A.J.M., S.S.Y. and A.P. supervised research. P.X.B, A.S. and A.P. wrote the manuscript. All authors contributed to manuscript review and editing. All authors have read and approved the final version of this manuscript and agree to be accountable for all aspects of the work in ensuring that questions related to the accuracy or integrity of any part of the work are appropriately investigated and resolved. All persons designated as authors qualify for authorship, and all those who qualify for authorship are listed.

## Funding

This work was funded by Vetenskapsrådet (The Swedish Research Council) grant 2022-00574 (A.P.), Hjärnfonden (The Swedish Brain Foundation) grants FO2024-0299 and FO2025-0364 (A.P.), and Region Östergötland hospital 'Från student till docent' stipend RÖ-1001370 (C.K.).

## Acknowledgements

We are grateful to the patients, their families and their doctors. We thank members of the Pantazis group for useful discussions; H. Peter Larsson for use of the OpusXpress TEVC robot; and members of the Elinder, Larsson, Liin and Pantazis groups for oocyte preparation. Flow cytometry experiments were performed using instrumentation at the Flow Cytometry unit of the Linköping University Core Facility: we are grateful to Mikael Pihl for expert consultation and support.

## Keywords

kv1.2, dee32, channelopathy, CMT1A, epilepsy, haploinsufficiency, ion channels, trafficking

## Supporting information

Additional supporting information can be found online in the Supporting Information section at the end of the HTML view of the article. Supporting information files available:

**Peer Review History**
**Supporting Information**

