## [Peer Review History · The Journal of Physiology]

Differential negative dominance by *KCNA2* variants associated with global developmental delay suggests *KCNA2* haploinsufficiency in humans

Pei Xin Boon, Amaia Jauregi-Miguel, Sümeyye Süheda Yasarbas, Serena Pozzi, Urban Karlsson, Ammar Husami, Charmaine Ko, Amelle Shillington, and Antonios Pantazis

DOI: 10.1113/JP290728

Corresponding author(s): Antonios Pantazis (antonios.pantazis@liu.se)

The following individual(s) involved in review of this submission have agreed to reveal their identity: Ken D O'Halloran (Referee #3)

Review Timeline:	Submission Date:	12-Dec-2025
	Editorial Decision:	10-Feb-2026
	Revision Received:	23-Feb-2026
	Editorial Decision:	10-Mar-2026
	Revision Received:	10-Mar-2026
	Accepted:	10-Mar-2026

Senior Editor: Eleonora Grandi

Reviewing Editor: Theodore Cummins

Transaction Report:

Re: JP-RP-2025-290728 "Differential negative dominance by two adjacent *KCNA2* variants associated with global developmental delay" by Pei Xin Boon, Amaia Jauregi-Miguel, Sümeyye Süheda Yasarbas, Serena Pozzi, Charmaine Ko, Amelle Shillington, and Antonios Pantazis

Dear Dr Pantazis,

Thank you for submitting your manuscript to The Journal of Physiology. It has been assessed by a Reviewing Editor and by 3 expert referees and we are pleased to tell you that it is potentially acceptable for publication following satisfactory major revision.

Please address all the points raised and incorporate all requested revisions or explain in your Response to Referees why a change has not been made. We hope you will find the comments helpful and that you will be able to return your revised manuscript within 2 months. If your article is NOT for a Special Issue, you may have 9 months to revise. If you require an extension, please contact journal staff: jp@physoc.org. Please note that this letter does not constitute a guarantee for acceptance of your revised manuscript.

REVISION CHECKLIST:

We look forward to receiving your revised submission.

Yours sincerely,

Eleonora Grandi
Senior Editor
The Journal of Physiology

REQUIRED ITEMS

- You must start the Methods section with a paragraph headed Ethical Approval. If experiments were conducted on humans, confirmation that informed consent was obtained, preferably in writing, that the studies conformed to the standards set by the latest revision of the Declaration of Helsinki and that the procedures were approved by a properly constituted ethics committee, which should be named, must be included in the article file. If the research study was registered (clause 35 of the Declaration of Helsinki), the registration database should be indicated, otherwise the lack of registration should be noted as an exception (e.g. The study conformed to the standards set by the Declaration of Helsinki, except for registration in a database). For further information see: <https://physoc.onlinelibrary.wiley.com/hub/human-experiments>.

- Please upload separate high-quality figure files via the submission form.

- Your paper contains Supporting Information of a type that we no longer publish, including supplementary tables and figures. Any information essential to an understanding of the paper must be included as part of the main manuscript and figures. The only Supporting Information that we publish are video and audio, 3D structures, program codes and large data files. Your revised paper will be returned to you if it does not adhere to our Supporting Information Guidelines.

- Papers must comply with the Statistics Policy: https://jp.msubmit.net/cgi-bin/main.plex?form_type=display_requirements#statistics.

In summary:

- If $n \leq 30$, all data points must be plotted in the figure in a way that reveals their range and distribution. A bar graph with data points overlaid, a box and whisker plot or a violin plot (preferably with data points included) are acceptable formats.

- If $n > 30$, then the entire raw dataset must be made available either as supporting information, or hosted on a not-for-profit repository, e.g. FigShare, with access details provided in the manuscript.

- 'n' clearly defined (e.g. x cells from y slices in z animals) in the Methods. Authors should be mindful of pseudoreplication.

- All relevant 'n' values must be clearly stated in the main text, figures and tables.

- The most appropriate summary statistic (e.g. mean or median and standard deviation) must be used. Standard Error of the Mean (SEM) alone is not permitted.

- Exact p values must be stated. Authors must not use 'greater than' or 'less than'. Exact p values must be stated to three

significant figures even when 'no statistical significance' is claimed.

EDITOR COMMENTS

Reviewing Editor:

Comments to the Author:

This is an interesting study that examines the impact of two different KCNA2 variants on Kv1.2 trafficking and functional properties. As mentioned by the reviewers, the data are intriguing, but the study is limited by the lack of functional evidence from neurons and limited details on the phenotype of the patient with the p.H310D variant. Additional data would be helpful. The reviewers raise other issues that should be addressed, including better consideration of the limitations of heterologous expression systems for predicting the functional consequences of channel mutations in neurons.

In the absence of data from neurons, the proposal that KCNA2 is haploinsufficient in humans seems preliminary. The conclusion should be softened and it should be clear that validation of speculation would involve confirming the impact of the p.G318D variant on trafficking in neurons.

Other minor issues:

Line 88: adjacent sites is misleading. The two mutations are not next to each other. They are in close proximity, but they are not adjacent. Please revise this sentence.

Line 152: "The rest of" please insert "the"

Line 157: Should cDNA be cRNA? Doesn't MessageMax T7 produce RNA, not DNA?

Line 203: Is SOS defined?

Line 283: Please clarify that 10 of the genes have been reported to harbor disease-associated variants "at this position".

Senior Editor:

Comments for Authors to ensure the paper complies with the Statistics Policy:

Please ensure that the paper complies with the Journal of Physiology Statistics policy.

Comments to the Author:

Thank you for submitting your study to The Journal of Physiology. Your manuscript was evaluated by two expert reviewers, a Reviewing Editor, and an Ethics Editor. While the reviewers recognized the potential interest of your work, they raised substantive concerns regarding the limited characterization of the patient carrying the p.H310D variant and the strength of the evidence supporting the conclusion that KCNA2 is a haploinsufficient gene in humans, resting solely on functional data obtained from heterologous expression systems. I concur with the Reviewing Editor's recommendation. Please also address the Ethics editor's request for additional detail.

REFEREE COMMENTS

Referee #1:

The work by Boon et al explores the effects of two KCNA2 variants (H310D, G318D) to the biophysical and trafficking properties of Kv1.2 channels. KCNA2 variants have been increasingly identified in patients with neurodevelopmental disorders. Despite this, the effects of different variants is not fully known. This work builds on the work done previously by the authors on H310 variants (PMID: 37883018) extending to a different variant and an additional neighboring amino acid (G318D) of the S4-S5 linker.

The data and experimental design is well conceived and done; although the absence of using auxiliary subunits or transporters (PMID: 30356053) limits the ability to make strong conclusions. The use of chimeras with Kv1.4 to bypass the trafficking deficits allowed the authors to better understand the biophysical effects of Kv1.2 variants; classifying them as

loss-of-function variants (reduced conductance and right shifted G-V). This was a clever approach.

Despite the several positives of this work, the overall conclusion that Kv1.2 variants H310D and G318D are likely acting as haploinsufficient variants is a preliminary conclusion. It's becoming increasingly clear that findings in heterologous cells such as *X. oocytes* or mammalian cells like CHO or HEK cells, do not always necessarily translate into neurons. In the absence of experiments using cultured neurons or mice expressing the variants any conclusions drawn about their effect in neurons is speculative. In the absence of more physiological experiments, the current work is incremental not providing conclusive mechanisms of how these variants might regulate neuronal excitability.

Referee #2:

In this study Boon and co-authors report the identification and functional characterization of two KCNA2 variants p.H310D and p.G318D, detected in patients with global developmental delay. Both variants introduce aspartate substitutions at adjacent, highly conserved residues within the Kv-channel superfamily. Functional analyses in *Xenopus laevis* oocytes and mammalian cells showed that both variants cause profound loss of channel function, abolishing potassium currents, disrupting surface trafficking, and impairing voltage-dependent activation. Tandem constructs pairing mutant Kv1.2 subunits with Kv1.4 displayed residual conductance with markedly altered voltage dependence, including right-shifted half-activation potentials. The two variants exhibited strikingly different dominant-negative properties. Variant p.H310D showed strong dominant-negative behavior, reducing current density to ~7% of wild-type levels in heterozygous conditions and severely limiting surface expression of wild-type subunits. In contrast, p.G318D displayed weaker dominant negativity, with ~32% residual conductance and largely preserved trafficking of wild-type subunits (~86%). The authors stated that: "Taken together with the p.G318D-patient's neurological symptoms, the latter suggests that KCNA2 is a haploinsufficient gene in humans." The study has a clear impact on the field, offers novel insight into underlying physiological mechanisms, and demonstrates originality. The experimental design is robust, and the data support valid and well-founded conclusions. Nevertheless, several issues should be addressed prior to publication.

Major concerns

1. The authors are encouraged to clarify the study's objectives and elaborate on how their key findings advance the understanding of the molecular and cellular mechanisms underlying developmental epileptic encephalopathy type 32. Doing so could also enable the formulation of a more impactful and representative title for the study.
2. The identification of a patient carrying a novel mutation is an important finding; however, it should be substantiated with additional clinical data (e.g., EEG recordings, etc.). A detailed correlation between the clinical features of this patient and those observed in individuals with similar KCNA2 mutations, as well as in patients with Charcot-Marie-Tooth disease type 1A (CMT1A) caused by inherited duplication of the PMP22 gene, would help clarify which symptoms are specifically attributable to the KCNA2 mutation. Furthermore, the authors state that the variant is *de novo* but do not provide the corresponding Sanger sequencing data electropherograms from the trio in the supplementary materials.
3. The p.H310D variant (NM_004974.4(KCNA2):c.928C>G) has been reported in ClinVar (Landrum et al., 2018; Variation ID: 1801703; Accession: VCV001801703.1) as a germline mutation identified in a patient with global developmental delay. Given that this variant demonstrates a strong dominant-negative effect-reducing current density to approximately 7% of wild-type levels under heterozygous conditions and markedly impairing the surface expression of wild-type subunits-efforts should be made, if possible, to collect additional clinical data, from the patient carrying this mutation, to assess whether it is associated with a more severe phenotype compared to the patient carrying the p.G318D mutation. This would contribute to refining the genotype-phenotype correlation, which remains an important open question.
4. Given that both mutations reside in a critical region potentially involved in interactions with the inactivation particle of Kv1.4, the authors should evaluate whether the Kv1.4-Kv1.2(H310D) and Kv1.4 Kv1.2(G318D) heteromers exhibit altered fast inactivation kinetics. This functional impairment may represent an additional mechanism contributing to the disease pathophysiology.
5. Particularly intriguing is the finding that inverting the subunit order in concatemers (i.e., Kv1.2-Kv1.4 vs Kv1.4-Kv1.2) completely prevents rescue of mutant channel expression. The authors are encouraged to discuss this phenomenon, potentially referencing studies on residues involved in channel trafficking, such as the following: Yang JW, Ulrich D, Andrasfalvy B, et al. Trafficking-dependent phosphorylation of Kv1.2 regulates voltage-gated potassium channel cell surface expression. *Proc Natl Acad Sci U S A.* 2007;104(50):20055-20060. doi:10.1073/pnas.0708574104. Also note that the N-type subunit Kv1.4 has a VXXSL motif on its C-terminus which, when altered, prevents surface trafficking and acts to retain the channel in the ER (Li et al., 2000). Mutation of this motif inhibits glycosylation. Alternatively, the pore of Kv1.4 also contains a forward-trafficking determinant, working by dictating the effect of N-glycosylation (Watanabe et al., 2004).

6. A seminal study by Servettini et al. (Proc Natl Acad Sci U S A. 2023 Aug;120(31):e2207978120. doi: 10.1073/pnas.2207978120), demonstrated that niflumic acid (NFA), an available anti-inflammatory drug, activates both Kv1.1 and Kv1.2 channels (see Supplementary Information). Their discussion highlighted that "KCNA2-related disorders caused by loss-of-function mutations in Kv1.2 could potentially benefit from NFA therapy", leaving this as an important open question. Testing NFA (e.g., 30-300 μ M) in their cellular expression systems (Figures 5-6) or in HEK293 cells-where Kv1.2 exhibits distinct "slow" and "fast" gating modes lost in oocytes (Baronas et al., J Neurosci. 2015;35:3515-3524)-would be highly informative. Demonstrating NFA efficacy could support its potential as a personalized therapy for KCNA2 loss-of-function mutations, offering some hope for improved quality of life in affected patients.

7. The heteromeric Kv1.2/Kv1.4 channels are prominently expressed along unmyelinated axons, axon initial segments, juxtaparanodes of myelinated axons, and presynaptic terminals. In the hippocampus, they localize to the middle molecular layer (perforant path axons synapsing onto dentate granule cells) and stratum lucidum of CA3 (mossy fiber axons) (JS Trimmer, KJ Rhodes Annu Rev Physiol 66, 477-519 (2004)). In the Discussion, the authors should further elaborate on how the functional defects associated with the identified mutations may affect brain physiology and contribute to the patient's clinical symptoms.

Referee #3:

Comments for Author:

Thank you for submitting your manuscript to The Journal of Physiology. Some additional details pertaining to ethics and animal welfare are required.

1. Line 99. Please confirm that the parents provided written informed consent for the use of their samples and that of 'the individual'. Please clarify that the individual was a child (legally a minor below the age limit for independent consent in the relevant jurisdiction), or elaborate if the circumstances were different (i.e. an incapacitated individual). Although one can infer from the text, please clarify that the consent process was provided for and approved by the IRB.

2. Line 116. For clarity, state that it was an absence of reflex responses to foot pinch that was observed. Provide details on what was determined.

3. Line 126. Although likely the same as before, nevertheless, provide details here of the anaesthetic agent used and how a state of deep anaesthesia was determined.

END OF COMMENTS

In this study Boon and *co-authors* report the identification and functional characterization of two *KCNA2* variants p.H310D and p.G318D, detected in patients with global developmental delay. Both variants introduce aspartate substitutions at adjacent, highly conserved residues within the Kv-channel superfamily. Functional analyses in *Xenopus laevis* oocytes and mammalian cells showed that both variants cause profound loss of channel function, abolishing potassium currents, disrupting surface trafficking, and impairing voltage-dependent activation. Tandem constructs pairing mutant Kv1.2 subunits with Kv1.4 displayed residual conductance with markedly altered voltage dependence, including right-shifted half-activation potentials. The two variants exhibited strikingly different dominant-negative properties. Variant p.H310D showed strong dominant-negative behavior, reducing current density to ~7% of wild-type levels in heterozygous conditions and severely limiting surface expression of wild-type subunits. In contrast, p.G318D displayed weaker dominant negativity, with ~32% residual conductance and largely preserved trafficking of wild-type subunits (~86%). The authors stated that: “Taken together with the p.G318D-patient’s neurological symptoms, the latter suggests that *KCNA2* is a haploinsufficient gene in humans.” The study has a clear impact on the field, offers novel insight into underlying physiological mechanisms, and demonstrates originality. The experimental design is robust, and the data support valid and well-founded conclusions. Nevertheless, several issues should be addressed prior to publication.

Major concerns

1. The authors are encouraged to clarify the study’s objectives and elaborate on how their key findings advance the understanding of the molecular and cellular mechanisms underlying developmental epileptic encephalopathy type 32. Doing so could also enable the formulation of a more impactful and representative title for the study.
2. The identification of a patient carrying a novel mutation is an important finding; however, it should be substantiated with additional clinical data (e.g., EEG recordings, etc.). A detailed correlation between the clinical features of this patient and those observed in individuals with similar *KCNA2* mutations, as well as in patients with *Charcot–Marie–Tooth* disease type 1A (CMT1A) caused by inherited duplication of the *PMP22* gene, would help clarify which symptoms are specifically attributable to the *KCNA2* mutation. Furthermore, the authors state that the variant is *de novo* but do not provide the corresponding Sanger sequencing data electropherograms from the trio in the supplementary materials.
3. The p.H310D variant (NM_004974.4(*KCNA2*):c.928C>G) has been reported in ClinVar (Landrum et al., 2018; Variation ID: 1801703; Accession: VCV001801703.1) as a germline mutation identified in a patient with global developmental delay. Given that this variant demonstrates a strong dominant-negative effect—reducing current density to approximately 7% of wild-type levels under heterozygous conditions and markedly impairing the surface expression of wild-type subunits—efforts should be made, if possible, to collect additional clinical data, from the patient carrying this mutation, to assess whether it is associated with a more severe phenotype compared to the patient carrying the p.G318D mutation. This would contribute to refining the genotype–phenotype correlation, which remains an important open question.
4. Given that both mutations reside in a critical region potentially involved in interactions with the inactivation particle of Kv1.4, the authors should evaluate whether the Kv1.4-Kv1.2(H310D) and Kv1.4-Kv1.2(G318D) heteromers exhibit altered fast inactivation kinetics. This functional impairment may represent an additional mechanism contributing to the disease pathophysiology.
5. Particularly intriguing is the finding that inverting the subunit order in concatemers (*i.e.*, Kv1.2-Kv1.4 vs Kv1.4-Kv1.2) completely prevents rescue of mutant channel expression. The authors are encouraged to discuss this phenomenon, potentially referencing studies on residues involved in channel trafficking, such as the following: Yang JW, Ulrich D, Andrasfalvy B, et al. Trafficking-dependent

phosphorylation of Kv1.2 regulates voltage-gated potassium channel cell surface expression. *Proc Natl Acad Sci U S A.* 2007;104(50):20055-20060. doi:10.1073/pnas.0708574104. Also note that the N-type subunit Kv1.4 has a VXXSL motif on its C-terminus which, when altered, prevents surface trafficking and acts to retain the channel in the ER (Li *et al.*, 2000). Mutation of this motif inhibits glycosylation. Alternatively, the pore of Kv1.4 also contains a forward-trafficking determinant, working by dictating the effect of N-glycosylation (Watanabe *et al.*, 2004).

6. A seminal study by Servettini *et al.* (*Proc Natl Acad Sci U S A.* 2023 Aug;120(31):e2207978120. doi: 10.1073/pnas.2207978120), demonstrated that niflumic acid (NFA), an available anti-inflammatory drug, activates both Kv1.1 and Kv1.2 channels (*see Supplementary Information*). Their discussion highlighted that “*KCNA2*-related disorders caused by loss-of-function mutations in Kv1.2 could potentially benefit from NFA therapy”, leaving this as an important open question. Testing NFA (e.g., 30–300 μ M) in their cellular expression systems (Figures 5–6) or in HEK293 cells—where Kv1.2 exhibits distinct “slow” and “fast” gating modes lost in oocytes (Baronas *et al.*, *J Neurosci.* 2015;35:3515–3524)—would be highly informative. Demonstrating NFA efficacy could support its potential as a personalized therapy for *KCNA2* loss-of-function mutations, offering some hope for improved quality of life in affected patients.
7. The heteromeric Kv1.2/Kv1.4 channels are prominently expressed along unmyelinated axons, axon initial segments, juxtaparanodes of myelinated axons, and presynaptic terminals. In the hippocampus, they localize to the middle molecular layer (perforant path axons synapsing onto dentate granule cells) and stratum lucidum of CA3 (mossy fiber axons) (JS Trimmer, KJ Rhodes *Annu Rev Physiol* **66**, 477–519 (2004)). In the Discussion, the authors should further elaborate on how the functional defects associated with the identified mutations may affect brain physiology and contribute to the patient’s clinical symptoms.

EDITOR COMMENTS

Reviewing Editor:

Comments to the Author:

This is an interesting study that examines the impact of two different KCNA2 variants on Kv1.2 trafficking and functional properties. As mentioned by the reviewers, the data are intriguing, but the study is limited by the lack of functional evidence from neurons and limited details on the phenotype of the patient with the p.H310D variant. Additional data would be helpful. The reviewers raise other issues that should be addressed, including better consideration of the limitations of heterologous expression systems for predicting the functional consequences of channel mutations in neurons.

In the absence of data from neurons, the proposal that KCNA2 is haploinsufficient in humans seems preliminary. The conclusion should be softened and it should be clear that validation of speculation would involve confirming the impact of the p.G318D variant on trafficking in neurons.

Thank you for reviewing our manuscript. We were careful to use soft language when it came to haploinsufficiency (“suggests”), but we have identified some places where we could soften it further, and we also added about the value of experiments in neurons.

Running title: “KCNA2 LOF variants & suggest haploinsufficiency”

Introduction (lines 77-78): “we hereby propose that KCNA2 is a haploinsufficient gene in humans” Changed to “it supports the premise”

Discussion (lines 471-472): “On these grounds, we propose that KCNA2 is haploinsufficient in humans.” Changed to: “Combined, these findings support the premise” and added the following statement:

“Yet while it is promising that the two variants showed differential negative dominance in diverse cellular contexts and culture conditions, we cannot exclude that p.G318D may exhibit strong negative dominance in the neurons of the developing human brain. Testing the impact of the p.G318D variant on trafficking in neurons would be an important step to support or disprove this premise.” (lines 472-476).

We also added, at the end of the Discussion (lines 534-535; when discussing the implications of KCNA2 haploinsufficiency for the efficacy of ASO treatment):

“Bearing in mind the caveat that p.G318D may be more negative-dominant in the human brain than in our experiments, we propose that the addition of K_v1-channel openers (e.g., (Borjesson et al., 2008; Liin et al., 2018; Silvera Ejneby et al., 2020; Manville et al., 2023; Servettini et al., 2023; Ronnelid & Elinder, 2024; Manville et al., 2025a; Manville et al.,

2025b)) in a combined ASO/pharmaceutical strategy, is likely to produce the most efficacious therapeutic outcome.”

Other minor issues:

Line 88: adjacent sites is misleading. The two mutations are not next to each other. They are in close proximity, but they are not adjacent. Please revise this sentence.

We agree, and replaced all instances of “adjacent” with “proximal”.

Line 152: "The rest of" please insert "the"

Done.

Line 157: Should cDNA be cRNA? Doesn't MessageMax T7 produce RNA, not DNA?

Correct, apologies for the mistake.

Line 203: Is SOS defined?

No, we now clarified it stands for “standard oocyte solution” (line 190).

Line 283: Please clarify that 10 of the genes have been reported to harbor disease-associated variants "at this position".

Done.

Senior Editor:

Comments for Authors to ensure the paper complies with the Statistics Policy:
Please ensure that the paper complies with the Journal of Physiology Statistics policy.

We have converted all our reported errors from SEM to SD.

Comments to the Author:

Thank you for submitting your study to The Journal of Physiology. Your manuscript was evaluated by two expert reviewers, a Reviewing Editor, and an Ethics Editor. While the reviewers recognized the potential interest of your work, they raised substantive concerns regarding the limited characterization of the patient carrying the p.H310D variant and the strength of the evidence supporting the conclusion that KCNA2 is a haploinsufficient gene in humans, resting solely on functional data obtained from heterologous expression systems. I concur with the Reviewing Editor's recommendation. Please also address the Ethics editor's request for additional detail.

Thank you for overseeing the review of our manuscript. We have addressed all concerns by the Reviewing Editor and the Reviewers, as well as the Ethics Editor. This includes performing additional experiments and providing more clinical information. We also revised our manuscript to make it more consistent with the J Physiol requirements: All errors are now reported as standard deviation. Revised text is marked by a grey highlight.

Despite repeated petitions to the geneticist responsible for the p.H310D variant deposition on ClinVar, we could not get more information about this patient: the family objected to reveal more, and we can but respect their privacy wishes.

Regarding Reviewer #2's concern about the phenotype-genotype correlation, we know (from ClinVar) that the patient suffers a neurological symptom associated with DEE32 (global developmental delay) and our work shows clear loss of function, both in terms of trafficking and voltage-dependent-opening, as well as strong evidence of negative dominance. The latter is agreement with the consensus in the field, that KCNA2 loss of function variants associate with DEE32. This information is salient to the phenotype-genotype correlation, since gain of KCNA2 function is also associated with DEE32. This implies that Kv1-channel openers will work better than blockers.

In addition, the p.H310D variant serves as a good "control", that our electrophysiological and trafficking experiments can discriminate between strong and weak negative dominance.

REFEREE COMMENTS

Referee #1:

Thank you for reviewing our work, and we are glad that you appreciated several aspects of it. Please find our response to your concerns below, in blue italics. Revised text in the manuscript is indicated by a grey highlight.

The work by Boon et al explores the effects of two KCNA2 variants (H310D, G318D) to the biophysical and trafficking properties of Kv1.2 channels. KCNA2 variants have been increasingly identified in patients with neurodevelopmental disorders. Despite this, the effects of different variants is not fully known. This work builds on the work done previously by the authors on H310 variants (PMID: 37883018) extending to a different variant and an additional neighboring amino acid (G318D) of the S4-S5 linker.

The data and experimental design is well conceived and done; although the absence of using auxiliary subunits or transporters (PMID: 30356053) limits the ability to make strong conclusions. The use of chimeras with Kv1.4 to bypass the trafficking deficits allowed the authors to better understand the biophysical effects of Kv1.2 variants; classifying them as loss-of-function variants (reduced conductance and right sifted G-V). This was a clever approach.

Yes, our experimental design is reduced in terms of molecular partners for Kv1.2 subunits; this was done on purpose, to extract the fundamental consequences of the mutations in the absence of other biological confounders. We note that the work on the Kv1.2-Slc7a5 which you refer (and of which we are big fans) was also performed in heterologous expression systems.

While auxiliary subunits, transporters and other partners may increase or decrease the overall trafficking of Kv1.2 subunits, we believe it is unlikely that they would affect the fundamental properties of the variants, as characterized in our work:

1) p.H310D causes loss of function in terms of both diminished trafficking and inhibited voltage-dependent opening; and it is strongly negative dominant

2) p.G318D cases loss of function in terms of both diminished trafficking and inhibited voltage-dependent opening; and is weakly negative dominant

3) the latter, combined with the p.G318D patient's neurological symptoms, is consistent with the premise that KCNA2 is haploinsufficient in humans.

It is very positive, that while not neuronal, both heterologous systems we used (amphibian oocytes cultured at 17 deg.C, and a primate cell line at 37 deg.C) do provide consistent results. I.e., our conclusions are valid over diverse cellular contexts and culture conditions.

Despite the several positives of this work, the overall conclusion that Kv1.2 variants H310D and G318D are likely acting as haploinsufficient variants is a preliminary conclusion. It's becoming increasingly clear that findings in heterologous cells such as X. oocytes or mammalian cells like CHO or HEK cells, do not always necessarily translate into neurons. In the absence of experiments using cultured neurons or mice expressing the variants any conclusions drawn about their effect in neurons is speculative. In the absence of more physiological experiments, the current work is incremental not providing conclusive mechanisms of how these variants might regulate neuronal excitability.

Thank you for sharing this view. Our work bears a necessary assumption, that Kv1.2 subunits in the human brain will recapitulate the properties we determine in our model systems. This assumption underpins all work on model systems, and any model system may “not always necessarily” recapitulate the real thing. We believe this is not a reason to dismiss any work on model systems as speculative. Yet one reason that supports the translatability of our work, is that our findings on KCNA2 loss-of-function are consistent with the pathological conditions in human patients, which provided the rationale to study these amino-acid substitutions in the first place.

Secondly, our conclusion on haploinsufficiency is the result of disproving our initial hypothesis: we expected that the two proximal, Asp-substitution variants would have the same effect on Kv1.2 function and trafficking. Our study looked very different on the outset! When this was not the case, as tested in both oocytes and monkey cells, the natural extrapolation of our results was the condition of haploinsufficiency. We agree that studies in neurons should be the next step, and we would be thrilled to discuss and even collaborate about disproving the new hypothesis (“KCNA2 is haploinsufficient”) at the neuronal level.

As for the work being incremental, we do respectfully disagree, because it provides the first evidence that KCNA2 is haploinsufficient in humans. Certainly, this evidence has the caveat that it comes from non-neuronal experimental models; whether it can be recapitulated under experimental conditions closer to those of the human brain remains to be investigated.

We were careful with the wording in our manuscript, disclosing our experimental systems as early as in the Key Points Summary and using soft language (“suggests...”, “we propose...”). In our revised version, we followed the Reviewing Editor’s recommendations, to soften the language further and mention trafficking experiments in neurons:

Running title: “KCNA2 LOF variants & suggest haploinsufficiency”

Introduction (lines 77-78): “we hereby propose that KCNA2 is a haploinsufficient gene in humans” Changed to “it supports the premise”

Discussion (lines 471-472): “On these grounds, we propose that KCNA2 is haploinsufficient in humans.” Changed to: “Combined, these findings support the premise” and added the following statement:

“Yet while it is promising that the two variants showed differential negative dominance in diverse cellular contexts and culture conditions, we cannot exclude that p.G318D may exhibit strong negative dominance in the neurons of the developing human brain. Testing the impact of the p.G318D variant on trafficking in neurons would be an important step to support or disprove this premise.” (lines 472-476).

We also added, at the end of the Discussion (lines 534-535; when discussing the implications of KCNA2 haploinsufficiency for the efficacy of ASO treatment):

“Bearing in mind the caveat that p.G318D may be more negative-dominant in the human brain than in our experiments, we propose that the addition of K_v1 -channel openers (e.g., (Borjesson et al., 2008; Liin et al., 2018; Silvera Ejneby et al., 2020; Manville et al., 2023; Servettini et al., 2023; Ronnelid & Elinder, 2024; Manville et al., 2025a; Manville et al., 2025b)) in a combined ASO/pharmaceutical strategy, is likely to produce the most efficacious therapeutic outcome.”

Referee #2:

Thank you for your positive appraisal of our work and your constructive comments. We have addressed all of your concerns, written below in blue italics. Changes to the text are indicated by a grey highlight.

In this study Boon and co-authors report the identification and functional characterization of two KCNA2 variants p.H310D and p.G318D, detected in patients with global developmental delay. Both variants introduce aspartate substitutions at adjacent, highly conserved residues within the Kv-channel superfamily. Functional analyses in *Xenopus laevis* oocytes and mammalian cells showed that both variants cause profound loss of channel function, abolishing potassium currents, disrupting surface trafficking, and impairing voltage-dependent activation. Tandem constructs pairing mutant Kv1.2 subunits with Kv1.4 displayed residual conductance with markedly altered voltage dependence, including right-shifted half-activation potentials. The two variants exhibited strikingly different dominant-negative properties. Variant p.H310D showed strong dominant-negative behavior, reducing current density to ~7% of wild-type levels in heterozygous conditions and severely limiting surface expression of wild-type subunits. In contrast, p.G318D displayed weaker dominant negativity, with ~32% residual conductance and largely preserved trafficking of wild-type subunits (~86%). The authors stated that: "Taken together with the p.G318D-patient's neurological symptoms, the latter suggests that KCNA2 is a haploinsufficient gene in humans." The study has a clear impact on the field, offers novel insight into underlying physiological mechanisms, and demonstrates originality. The experimental design is robust, and the data support valid and well-founded conclusions. Nevertheless, several issues should be addressed prior to publication.

Major concerns

1. The authors are encouraged to clarify the study's objectives and elaborate on how their key findings advance the understanding of the molecular and cellular mechanisms underlying developmental epileptic encephalopathy type 32. Doing so could also enable the formulation of a more impactful and representative title for the study.

Thank you for urging us to clarify our work. We have modified our text in the following ways:

1. At the end of the introduction (lines 70-72), we added:

"Our objectives were to characterize the consequences of the variants for the channel

functional properties and subunit trafficking; as well as evaluate the interaction of wild-type and variant subunits, emulating the heterozygous condition.”

2. We agree that the title does not fully capture the implications of our work. We changed it, from:

“Differential negative dominance by two adjacent KCNA2 variants associated with global developmental delay”

To:

“Differential negative dominance by ~~two adjacent~~ KCNA2 variants associated with global developmental delay ~~suggests KCNA2 haploinsufficiency~~”

2. The identification of a patient carrying a novel mutation is an important finding; however, it should be substantiated with additional clinical data (e.g., EEG recordings, etc.). A detailed correlation between the clinical features of this patient and those observed in individuals with similar KCNA2 mutations, as well as in patients with Charcot-Marie-Tooth disease type 1A (CMT1A) caused by inherited duplication of the PMP22 gene, would help clarify which symptoms are specifically attributable to the KCNA2 mutation. Furthermore, the authors state that the variant is de novo but do not provide the corresponding Sanger sequencing data electropherograms from the trio in the supplementary materials.

CMT1A presents with peripheral deficits (motor weakness) and tends to develop later in life due to demyelination. KCNA2 variants have a broad symptomatology. The patient’s central symptoms (febrile seizures, generalized tonic-clonic seizures and borderline intellectual disability) are generally consistent with symptoms observed in other patients with KCNA2 variants. In our revised manuscript, we write (lines 282-291):

“Her history includes febrile seizures and two generalized tonic-clonic seizures. She also has cortical vision abnormalities, attention-deficit/hyperactivity disorder, staring spells, and borderline intellectual disability (intelligence quotient 76)” ... “CMT1A does not present this early, and the duplication does not explain the congenital or CNS abnormalities, so the KCNA2 variant is proposed to underlie the early-onset, central-nervous-system pathology—as has been reported in other patients with KCNA2 variants (Syrbe et al., 2015; Masnada et al., 2017; Doring et al., 2021).”

We have added sequencing data proving the de novo emergence of the mutation in Supplementary figure 1. One EEG was recorded in the sleeping patient; while no seizures were recorded at that time, the attending physician commented that the EEG was “abnormal”, with “potentially epileptiform” activity. This information is included in the Appendix.

3. The p.H310D variant (NM_004974.4(KCNA2):c.928C>G) has been reported in ClinVar (Landrum et al., 2018; Variation ID: 1801703; Accession: VCV001801703.1) as a germline mutation identified in a patient with global developmental delay. Given that this variant demonstrates a strong dominant-negative effect-reducing current density to approximately 7% of wild-type levels under heterozygous conditions and markedly impairing the surface expression of wild-type subunits-efforts should be made, if possible, to collect additional clinical data, from the patient carrying this mutation, to assess whether it is associated with a more severe phenotype compared to the patient carrying the p.G318D mutation. This would contribute to refining the genotype-phenotype correlation, which remains an important open question.

We agree that this would have been very valuable information, and we reached out to the Ruhr University Bochum Genomics Core Facility for more information. Unfortunately, the health professional who deposited the variant to ClinVar responded that the patient's parents do not want to disclose more information. We can only respect their wishes.

A propos the phenotype-genotype correlation, we know (from ClinVar) that the patient suffers a neurological symptom associated with DEE32 (global developmental delay) and our work shows clear loss of function, both in terms of trafficking and voltage-dependent-opening, as well as strong evidence of negative dominance. The latter is agreement with the consensus in the field, that KCNA2 loss of function variants associate with DEE32. This information is salient to the phenotype-genotype correlation, since gain of KCNA2 function is also associated with DEE32. This implies that Kv1-channel openers will work better than blockers.

In addition, the p.H310D variant serves as a good "control", that our electrophysiological, trafficking and pharmacology experiments can discriminate between strong and weak negative dominance.

4. Given that both mutations reside in a critical region potentially involved in interactions with the inactivation particle of Kv1.4, the authors should evaluate whether the Kv1.4-Kv1.2(H310D) and Kv1.4 Kv1.2(G318D) heteromers exhibit altered fast inactivation kinetics. This functional impairment may represent an additional mechanism contributing to the disease pathophysiology.

*This is an important question. We analyzed the extent of inactivation, taking the ratio of current at the end of the 100-ms pulse (I_{100}) over peak current (I_{peak}). The dimeric constructs with variant Kv1.2 subunits had a tendency to inactivate more, but this effect was not significant. This analysis is included in the **new panel E, in figure 3**. Perhaps the existence of wild-type Kv1.4 subunits (with unaffected particle interaction sites) is sufficient to confer full inactivation, despite the presence of variant Kv1.2 subunits.*

5. Particularly intriguing is the finding that inverting the subunit order in concatemers (i.e., Kv1.2-Kv1.4 vs Kv1.4-Kv1.2) completely prevents rescue of mutant channel expression. The authors are encouraged to discuss this phenomenon, potentially referencing studies on residues involved in channel trafficking, such as the following: Yang JW, Ulrich D, Andrasfalvy B, et al. Trafficking-dependent phosphorylation of Kv1.2 regulates voltage-gated potassium channel cell surface expression. Proc Natl Acad Sci U S A. 2007;104(50):20055-20060. doi:10.1073/pnas.0708574104. Also note that the N-type subunit Kv1.4 has a VXXSL motif on its C-terminus which, when altered, prevents surface trafficking and acts to retain the channel in the ER (Li et al., 2000). Mutation of this motif inhibits glycosylation. Alternatively, the pore of Kv1.4 also contains a forward-traffic determinant, working by dictating the effect of N-glycosylation (Watanabe et al., 2004).

We have added the following part in the revised manuscript (lines 345-354):

“This result, that Kv1.2/Kv1.4 dimeric constructs exhibit less functional expression when the Kv1.2-variant partner is N-terminal, was also observed in other trafficking-deficient variants (p.F233S, p.H310R) (Nilsson et al., 2022; Minguez-Vinas et al., 2023). Post-translational modifications, like phosphorylation and glycosylation, can affect Kv1.2 and Kv1.4-subunit trafficking (Li et al., 2000; Watanabe et al., 2004; Yang et al., 2007), although these are presumably preserved when Kv1.2 is either N- or C-terminal. Our result here suggests that (i) a trafficking defect occurs early in biosynthesis, e.g., during translation or ER translocation; and (ii) constructs with an N-terminal Kv1.4 partner may traffic better due to cooperative translocation.”

6. A seminal study by Servettini et al. (Proc Natl Acad Sci U S A. 2023 Aug;120(31):e2207978120. doi: 10.1073/pnas.2207978120), demonstrated that niflumic acid (NFA), an available anti-inflammatory drug, activates both Kv1.1 and Kv1.2 channels (see Supplementary Information). Their discussion highlighted that "KCNA2-related disorders caused by loss-of-function mutations in Kv1.2 could potentially benefit from NFA therapy", leaving this as an important open question. Testing NFA (e.g., 30-300 μ M) in their cellular expression systems (Figures 5-6) or in HEK293 cells-where Kv1.2 exhibits distinct "slow" and "fast" gating modes lost in oocytes (Baronas et al., J Neurosci. 2015;35:3515-3524)-would be highly informative. Demonstrating NFA efficacy could support its potential as a personalized therapy for KCNA2 loss-of-function mutations, offering some hope for improved quality of life in affected patients.

This is an interesting idea. We have performed the experiments, trialing 300 μ M NFA perfusion in our oocyte expression system. They show a consistent left shift in the voltage dependence and increase of current in the heterozygous condition for both variants. At -20 mV, the current is increased by 3-fold for all conditions. We have revised

our Methods (lines 212-218) and the Results section (lines 403-411), and introduced new figure 7.

In addition, we mention as closing words to the article (lines 535-539):

“we propose that the addition of Kv1-channel openers (e.g., (Borjesson et al., 2008; Liin et al., 2018; Silvera Ejneby et al., 2020; Manville et al., 2023; Servettini et al., 2023; Ronnelid & Elinder, 2024; Manville et al., 2025a; Manville et al., 2025b)) in a combined ASO/pharmaceutical strategy, is likely to produce the most efficacious therapeutic outcome.”

7. The heteromeric Kv1.2/Kv1.4 channels are prominently expressed along unmyelinated axons, axon initial segments, juxtaparanodes of myelinated axons, and presynaptic terminals. In the hippocampus, they localize to the middle molecular layer (perforant path axons synapsing onto dentate granule cells) and stratum lucidum of CA3 (mossy fiber axons) (JS Trimmer, KJ Rhodes *Annu Rev Physiol* 66, 477-519 (2004)). In the Discussion, the authors should further elaborate on how the functional defects associated with the identified mutations may affect brain physiology and contribute to the patient's clinical symptoms.

Thank you for urging us to elaborate on the clinical effects of our variants and suggesting the interaction of Kv1.2/Kv1.4 channels in the hippocampus. We included a limited discussion, to also be in line with the comments from the Reviewing Editor and Reviewer #1, that our work in heterologous expression systems has limited translatability to neurons (lines 442-451).

“Our work on heterologous expression systems in the absence of neuronal $K_v1.2$ molecular partners is too removed from the context of the human brain for a definitive discussion on epilepsy and developmental delay. We speculate that loss of $K_v1.2$ function would broaden the action potentials of both excitatory and inhibitory neurons. In excitatory synapses, this would increase synaptic release (Kole et al., 2007). Yet in inhibitory neurons, fast-gating K_v3 channels (Sekirnjak et al., 1997; Erisir et al., 1999; Tasic et al., 2016), would act to normalize action potential duration (Rowan et al., 2014; Labro et al., 2015); in this way, inhibitory synaptic release would be preserved. The resulting imbalance of excitatory and inhibitory transmission could thus promote epileptogenesis (McCormick & Contreras, 2001). ”

Referee #3:

Comments for Author:

Thank you for submitting your manuscript to The Journal of Physiology. Some additional details pertaining to ethics and animal welfare are required.

Thank you for ensuring our ethical research conduct is appropriately reported. Please find our response to your concerns below in blue italics. Changes to the text in the revised manuscript are indicated by a vertical line on the left margin.

1. Line 99. Please confirm that the parents provided written informed consent for the use of their samples and that of 'the individual'. Please clarify that the individual was a child (legally a minor below the age limit for independent consent in the relevant jurisdiction), or elaborate if the circumstances were different (i.e. an incapacitated individual). Although one can infer from the text, please clarify that the consent process was provided for and approved by the IRB.

The patient is a minor (now included in the methods), and her parents consented to participation in research and publication. They provided written consent and are enrolled under our IRB at Cincinnati Children's #2013-7327.

2. Line 116. For clarity, state that it was an absence of reflex responses to foot pinch that was observed. Provide details on what was determined.

You are right, it was the absence of reflex responses to foot pinch. Now included in the methods.

3. Line 126. Although likely the same as before, nevertheless, provide details here of the anaesthetic agent used and how a state of deep anaesthesia was determined.

Correct, it was the same procedure as above. Now included in the Methods.

END OF COMMENTS

Dear Dr Pantazis,

Re: JP-RP-2026-290728R1 "Differential negative dominance by *KCNA2* variants associated with global developmental delay suggests *KCNA2* haploinsufficiency in humans" by Pei Xin Boon, Amaia Jauregi-Miguel, Sümeyye Süheda Yasarbas, Serena Pozzi, Urban Karlsson, Ammar Husami, Charmaine Ko, Amelle Shillington, and Antonios Pantazis

Thank you for submitting your manuscript to The Journal of Physiology. It has been assessed by a Reviewing Editor and by 3 expert referees and we are pleased to tell you that it is acceptable for publication following satisfactory minor revision.

LANGUAGE EDITING AND SUPPORT FOR PUBLICATION: If you would like help with English language editing, or other article preparation support, Wiley Editing Services offers expert help, including English Language Editing, as well as translation, manuscript formatting, and figure formatting at www.wileyauthors.com/eoo/preparation. You can also find resources for Preparing Your Article for general guidance about writing and preparing your manuscript at www.wileyauthors.com/eoo/prepresources.

REVISION CHECKLIST:

Please upload two versions of your manuscript text: one with all relevant changes highlighted and one clean version with no changes tracked. The manuscript file should include all tables and figure legends, but each figure/graph should be uploaded as separate, high-resolution files. The journal is now integrated with Wiley's Image Checking service. For further details, see: <https://www.wiley.com/en-us/network/publishing/research-publishing/trending-stories/upholding-image-integrity-wileys->

image-screening-service

We look forward to receiving your revised submission.

Yours sincerely,

Eleonora Grandi
Senior Editor
The Journal of Physiology

REQUIRED ITEMS

- You must start the Methods section with a paragraph headed Ethical Approval. If experiments were conducted on humans, confirmation that informed consent was obtained, preferably in writing, that the studies conformed to the standards set by the latest revision of the Declaration of Helsinki and that the procedures were approved by a properly constituted ethics committee, which should be named, must be included in the article file. If the research study was registered (clause 35 of the Declaration of Helsinki), the registration database should be indicated, otherwise the lack of registration should be noted as an exception (e.g. The study conformed to the standards set by the Declaration of Helsinki, except for registration in a database). For further information see: <https://physoc.onlinelibrary.wiley.com/hub/human-experiments>.

EDITOR COMMENTS

Reviewing Editor:

Per the Ethics editor, the following is needed:

"Line 84: You must state that the study conformed to the principles of the Declaration of Helsinki. If the study was registered as a trial include details. If not, state that the study conformed to the principles of the Declaration of Helsinki, except for registration in a public database."

Comments to the Author (Required):

The authors have done a reasonable job revising the manuscript. Experiments in neurons would have been great, but the paper makes a solid contribution and presents a hypothesis that can be followed up on. If the authors could address the final issue raised by ethics editor regarding the principles of the Declaration of Helsinki, we can move forward.

Senior Editor:

Thank you for revising the paper. Please address the remaining comment by the Ethics Editor.

REFEREE COMMENTS

Referee #1:

This is a revised manuscript testing the effects of Kv1.2 mutations identified in patients with severe neurodevelopment disorders. As I discussed before, the quality of the work is exceptional, but the lack of use of neurons to test the hypothesis of haploinsufficiency substantially reduces the impact of this work. I appreciate the authors attempt to tone down their conclusions, but the conclusions are still speculative.

Referee #2:

No further comments.

Referee #3 (ethics review):

Thank you for making revisions to the text based on my previous comments. All points raised were satisfactorily addressed. My sincere apologies for not raising this in the previous round of comments but there is one remaining revision required.

Line 84: You must state that the study conformed to the principles of the Declaration of Helsinki. If the study was registered as a trial include details. If not, state that the study conformed to the principles of the Declaration of Helsinki, except for registration in a public database.

END OF COMMENTS

Review of Revised Manuscript for Journal of Physiology (London)

Manuscript n. JP-RP-2026-290728R1

Dear Editor,

Thank you for the opportunity to review the revised version of this manuscript.

The authors have satisfactorily addressed all of the concerns and comments I raised in my previous review. The revisions are thorough, scientifically sound, and have significantly strengthened the clarity and rigor of the manuscript. I have no further comments at this stage.

In my opinion, the manuscript is now potentially acceptable for publication in the *Journal of Physiology (London)*.

Yours sincerely,

Professor Mauro Pessia

EDITOR COMMENTS

Reviewing Editor:

Per the Ethics editor, the following is needed:

"Line 84: You must state that the study conformed to the principles of the Declaration of Helsinki. If the study was registered as a trial include details. If not, state that the study conformed to the principles of the Declaration of Helsinki, except for registration in a public database."

Comments to the Author (Required):

The authors have done a reasonable job revising the manuscript. Experiments in neurons would have been great, but the paper makes a solid contribution and presents a hypothesis that can be followed up on. If the authors could address the final issue raised by ethics editor regarding the principles of the Declaration of Helsinki, we can move forward.

Thank you, we made the addition in our Methods: In p.4, lines 112-113 we now mention: "The study conformed to the principles of the Declaration of Helsinki, except for registration in a public database."

We also took the opportunity to improve the writing in the First Author Profile.

Senior Editor:

Thank you for revising the paper. Please address the remaining comment by the Ethics Editor.

Thank you, we made the addition in our Methods: In p.4, lines 112-113 we now mention: "The study conformed to the principles of the Declaration of Helsinki, except for registration in a public database."

We also took the opportunity to improve the writing in the First Author Profile.

REFEREE COMMENTS

Referee #1:

This is a revised manuscript testing the effects of Kv1.2 mutations identified in patients with severe neurodevelopment disorders. As I discussed before, the quality of the work is exceptional, but the lack of use of neurons to test the hypothesis of haploinsufficiency substantially reduces the impact of this work. I appreciate the authors attempt to tone down their conclusions, but the conclusions are still speculative.

Thank you for your positive comments.

Referee #2:

No further comments.

Thank you.

Referee #3 (ethics review):

Thank you for making revisions to the text based on my previous comments. All points raised were satisfactorily addressed. My sincere apologies for not raising this in the previous round of comments but there is one remaining revision required.

Line 84: You must state that the study conformed to the principles of the Declaration of Helsinki. If the study was registered as a trial include details. If not, state that the study conformed to the principles of the Declaration of Helsinki, except for registration in a public database.

In p.4, lines 112-113 we now mention: "The study conformed to the principles of the Declaration of Helsinki, except for registration in a public database."

END OF COMMENTS

Dear Professor Pantazis,

Re: JP-RP-2026-290728R2 "Differential negative dominance by *KCNA2* variants associated with global developmental delay suggests *KCNA2* haploinsufficiency in humans" by Pei Xin Boon, Amaia Jauregi-Miguel, Sümeyye Süheda Yasarbas, Serena Pozzi, Urban Karlsson, Ammar Husami, Charmaine Ko, Amelle Shillington, and Antonios Pantazis

We are pleased to tell you that your paper has been accepted for publication in The Journal of Physiology.

Yours sincerely,

Eleonora Grandi
Senior Editor
The Journal of Physiology

IMPORTANT POINTS TO NOTE FOLLOWING ACCEPTANCE OF YOUR PAPER:

- **IMPORTANT NOTICE ABOUT OPEN ACCESS:** To assist authors whose funding agencies mandate immediate public access to published research findings, The Journal of Physiology allows authors to pay an Open Access (OA) fee to have their papers made freely available immediately on publication.

The Corresponding Author will receive an email from Wiley with details on how to register or log in to Wiley Authors where you will be able to place an order.

- You can check if your funder or institution has a Wiley Open Access Account here:
<https://authors.wiley.com/author-resources/Journal-Authors/open-access/author-compliance-tool.html>

- You can help your research get the attention it deserves! Check out Wiley's free Promotion Guide for best-practice recommendations for promoting your work at: www.wileyauthors.com/eeo/guide. You can learn more about Wiley Editing Services which offers professional video, design, and writing services to create shareable video abstracts, infographics, conference posters, lay summaries, and research news stories for your research at: www.wileyauthors.com/eeo/promotion.

- If you would like to receive our 'Research Roundup', a monthly newsletter highlighting the cutting-edge research published in The Physiological Society's family of journals (The Journal of Physiology, Experimental Physiology, Physiological Reports, The Journal of Nutritional Physiology and The Journal of Precision Medicine: Health and Disease), please click this link, fill in your name and email address and select 'Research Roundup':
<https://www.physoc.org/journals-and-media/membernews>

EDITOR COMMENTS

Reviewing Editor:

Revised appropriately. Thank you for your submission.

Senior Editor:

Congratulations!